# Enrichment of organic nitrogen in fog residuals observed in the Italian Po Valley

Fredrik Mattsson[a,b], Almuth Neuberger[a,b], Liine Heikkinen[a,b], Yvette Gramlich[a,b,c], Marco Paglione[d], Matteo Rinaldi[d], Stefano Decesari[d], Paul Zieger[a,b], Ilona Riipinen[a,b], and Claudia Mohr[a,b,c,e]

[a]Department of Environmental Science, Stockholm University, Stockholm, Sweden
[b]Bolin Centre for Climate Research, Stockholm, Sweden
[c]PSI Center for Energy and Environmental Sciences, 5232 Villigen PSI, Switzerland
[d]Institute of Atmospheric Sciences and Climate, National Research Council, Bologna, Italy
[e]Department of Environmental Systems Science, ETH Zurich, Zürich, Switzerland

**Correspondence:** Claudia Mohr (claudia.mohr@psi.ch)

**Abstract.** While aerosol-cloud interactions have been extensively investigated, large knowledge gaps still exist. Atmospheric organic nitrogen (ON) species and their formation in the aqueous phase are potentially important due to their influence on aerosol optical and hygroscopic properties, and their adverse effects on human health. This study aimed to characterize the wintertime aerosol and fog chemical composition, with focus on the formation of ON, at a rural site in the Italian Po Valley.

Online chemical characterization of interstitial aerosol (non-activated particles) and fog residuals (dried fog droplets) were performed in parallel. Fog residuals were sampled using a Ground-based Counterflow Virtual Impactor (GCVI) inlet and analyzed by a Soot-Particle Aerosol Mass Spectrometer (SP-AMS), while the interstitial aerosol was characterized by a High-Resolution Time of Flight AMS (HR-TOF-AMS). Our results revealed an enhancement of nitrate ($NO_3^-$; 43.3 % vs. 34.6 %), ammonium ($NH_4^+$; 15.2 % vs. 11.7 %), and sulfate ($SO_4^{2-}$; 10.5 % vs. 6.6 %) in the fog residuals compared to the ambient

non-fog aerosol, while the organic aerosol (OA; 27.6 % vs. 39.4 %) and refractory black carbon (rBC; 2.3 % vs. 6.3 %) were less abundant. An enrichment of ON was observed in the fog, mainly consisting of $C_xH_yN_1^+$ ions, partly originating from amines. $C_xH_yN_2^+$ ions, fragments linked to imidazoles were over proportionally present in the fog, suggesting aqueous-phase formation, which was verified by proton nuclear magnetic resonance ($^1$H-NMR) spectroscopy. This study demonstrates that fogs and clouds are potentially important sinks for gaseous nitrogen species and media for aqueous production of nitrogen-

containing organic aerosol in the atmosphere.

# 1 Introduction

Aerosol particles have a significant impact on clouds and fogs, and a substantial indirect climate effect as a result (Szopa et al., 2021). The complex aerosol-cloud interactions are poorly constrained, especially on the micro-physical scale (Masson-Delmotte et al., 2021). The interactions of aerosol particles with fogs are largely analogous to the interactions with higher-altitude clouds. Fog droplets form by the condensation of water vapor onto aerosol particles, which act as cloud condensation nuclei (CCN). For radiation fog, this typically occurs overnight as the near-surface air cools and becomes saturated with respect to water vapor. The physical and chemical properties of the activated CCN influence the properties of the fog, such as the number and size of the fog droplets, the fog life cycle, as well as its effect on Earth's radiation budget (Jia et al., 2019; Duplessis et al., 2021).

Fog droplets can act as reactors for transforming both gases and aerosol particles through aqueous-phase reactions. For aerosol particles, this may result in an altered size, shape, and chemical composition of fog residuals as the fog dissipates (Ge et al., 2012; Seinfeld and Pandis, 2016). The organic aerosol (OA), which overall constitutes a significant mass fraction (20-80 %) of the atmospheric particulate matter (Zhang et al., 2007), has been shown to undergo chemical transformation in fog water (Mazzoleni et al., 2010; Ehrenhauser et al., 2012), with possible implications for aerosol reactivity and toxicity. Studying OA in the atmosphere poses significant challenges due to the vast range of different compounds (Goldstein and Galbally, 2007), as well as the continuous evolution as these compounds reside in the atmosphere (Jimenez et al., 2009). A large, often dominant fraction of this atmospheric OA is considered secondary organic aerosol (SOA) (Hallquist et al., 2009). Volatile organic compounds (VOCs) are SOA precursors, which can be emitted from both biogenic and anthropogenic sources. A major pathway for SOA formation is the partitioning of VOC-oxidation products into the particle phase. This process can be enhanced at high relative humidity for water-soluble compounds, where cloud/fog droplets or deliquesced aerosol particles act as a reactive media for new aqueous SOA (aqSOA) (Ervens et al., 2011).

While the formation mechanisms, precursors involved, and the necessary conditions for aqSOA production are poorly understood, key reactions involved are recognized. Oxidation reactions occurring in the aqueous phase play a significant role in generating low-volatility products, such as dicarboxylic acids or keto acids (Ervens, 2015). Another pathway for certain VOCs, such as ethene and isoprene, to contribute to aqSOA is through oxidation leading to the formation of glyoxal ($C_2H_2O_2$) or methylglyoxal ($C_3H_4O_2$), which can generate low-volatility species by hydration or oligomerization reactions (Herrmann et al., 2015). For example, in the presence of amines or amino acids, dissolved glyoxal or methylglyoxal can form nitrogen-containing compounds, such as imidazole (De Haan et al., 2011; Ervens et al., 2011). These oxidized products result in an enhanced particulate mass, which can remain in the particle phase after fog/cloud dissipation (Hallquist et al., 2009). Phenolic compounds, such as syringol and guaiacol have been found to form dimers or oligomers upon oxidation in the aqueous phase (Sun et al., 2010; Yu et al., 2014).

Atmospheric organic nitrogen (ON) are compounds ubiquitously observed in the atmosphere, and known to partake in aqueous processes in fog and cloud droplets (Zhang and Anastasio, 2001; Laskin et al., 2015). However, their formation and reactions involved are poorly understood. Gaseous and particulate ON represents about 30 % of the total atmospheric reactive

nitrogen and plays an important role in the terrestrial and aquatic nitrogen cycles (Cape et al., 2011; Jickells et al., 2013). Sources of ON include emissions from vehicle exhaust, biomass burning, and agriculture (Rabaud et al., 2003; Cape et al., 2011), while secondary formation may occur through VOC oxidation by the nitrate radical (Ng et al., 2017), and aqueous-phase reactions in fogs or clouds (Rehbein et al., 2011). ON includes a large range of compounds, of which amines, amino acids, amides, organo-nitrates, nitro-compounds, and imidazoles (N-heterocyclic compounds) have been observed in fog or

cloud water (Zhang et al., 2012; Youn et al., 2015; Chen et al., 2018; Kim et al., 2019; Ge et al., 2024). ON can affect the climate, as it can influence both optical (Laskin et al., 2015; Chen et al., 2018) and hygroscopic properties (Powelson et al., 2014) of the aerosol particles – thereby influencing both the direct and indirect ways aerosol particles interact with radiation. Moreover, ON species, such as amines and imidazoles, can have adverse effects on the environment and human health (Ge et al., 2011).

The chemical analysis of fog water and the interstitial aerosol particles has proven to be an efficient way to assess the importance of aqSOA formation in cloud water (Gilardoni et al., 2016; Kim et al., 2019; Paglione et al., 2020). Within Europe, the Po Valley stands out as an important measurement site, where the fog-processing of agricultural and industrial emissions has been studied. With a population of approximately 20 million people, the Po Valley is a densely populated region, with one of the highest levels of air pollution in Europe, recording annual mean $PM_{2.5}$ (particulate matter with a diameter of 2.5 $\mu$m or smaller)

concentrations between 20 - 30 $\mu$g m$^{-3}$ at several sites, well above the World Health Organization's recommended guidelines of 5 $\mu$g m$^{-3}$ (EEA, 2020). The emission sources are mainly industrial, agricultural, and residential biomass burning (Scotto et al., 2021). The valley is surrounded by the Alps in the north and west, and the Apennines in the south. This often results in a stagnant atmosphere, a compact mixing layer, and weak wind speeds (Vecchi et al., 2019). As a result, long-lasting episodes of high surface relative humidity (RH) and radiation fog events frequently occur during the colder winter months (Giulianelli

et al., 2014). These conditions prevent the dispersion of air pollution, which can lead to the formation and build-up of secondary aerosol particles (Perrino et al., 2014). Over the past three decades, numerous research studies have delved into the Po Valley fog. For example, the fog water and aerosol acidity have exhibited opposite trends. The fog water pH has increased from about 5.5 to 6.5, likely due to the decreasing trends of nitrate ($NO_3^-$) and sulfate ($SO_4^{2-}$) (Giulianelli et al., 2014). Simultaneously, the aerosol pH decreased from about 5 to 4, which was mainly contributed to total ammonium ($NH_4^+$), i.e. the buffering effect of

semi-volatile ammonia ($NH_3$), while decreasing RH and increasing temperature in the region also were major drivers (Paglione et al., 2021; Weber et al., 2016). The physical properties of the Po Valley aerosol and fog droplets have been investigated by e.g. Noone et al. (1992), who found that between 20 to 30 % of the accumulation mode aerosol mass was scavenged by the fog. Biomass burning emissions have a significant influence on the aerosol mass, especially during the cold months, and contribute to aqSOA formation (Gilardoni et al., 2016; Paglione et al., 2020). Furthermore, it has been established that SOA dominates the

OA mass in this region, where biomass burning is a major contributor (30 - 80 %) to the $PM_1$ mass, both to the primary OA and the SOA during the winter months (Saarikoski et al., 2012; Paglione et al., 2020). ON has previously been observed in the Po Valley. For example, Montero-Martínez et al. (2014) found a good linear correlation between water soluble ON and $NO_3^-$, $NH_4^+$ and $SO_4^{2-}$ ions, while biomass burning was proven to be a potentially significant night-time contributor to ON. Saarikoski et al. (2012) identified a nitrogen-containing OA factor from positive matrix factorization (PMF). $C_xH_yN_1^+$ compounds comprised

14 % of this factor, which also showed a good linear correlation with $NO_3^-$ and $NH_4^+$. The role of fog in ON formation and content of aerosols in the Po Valley has not been investigated in detail to date.

There are many challenges involved in obtaining in-situ cloud measurements, as it typically requires an aircraft. By focusing on fog, many of these restrictions can be avoided. The analysis of aerosol-fog interactions has so far been restricted to offline chemical analysis of fog water or aerosol chemical composition, or to in-situ analysis of interstitial aerosol particles. In situ-measurements of fog droplet residuals are lacking, limiting our understanding of aerosol-fog interactions.

In this study, the Po Valley fog was thoroughly investigated by performing online measurements of the ambient aerosol and the fog residuals, as well as collecting fog water for offline analysis, as part of the Fog and Aerosol InteRAction Research Italy (FAIRARI, Neuberger et al., 2025) campaign 2021/2022. The composition of the ambient aerosol, interstitial aerosol, and fog residuals was characterized using a Soot-Particle Aerosol Mass Spectrometer (SP-AMS) and a High-Resolution Time of Flight Aerosol Mass Spectrometer (HR-TOF-AMS) in parallel, while fog water samples were studied offline using the SP-AMS and a Proton Nuclear Magnetic Resonance ($^1$H-NMR) spectroscopy. Previous studies performing online quantification of ON exist (e.g. Kim et al., 2019; Ge et al., 2024; Graeffe et al., 2022) but are relatively few compared to studies on other OA species. This work presents an effort to highlight particulate ON existing in fog water and fog residuals.

## 2 Methods

### 2.1 The FAIRARI campaign

The Fog and Aerosol InteRAction Research Italy (FAIRARI) campaign was conducted between November 2021 and April 2022, and took place at San Pietro Capofiume (SPC) in the Po Valley, Italy (Neuberger et al., 2025). The main goals of the campaign were to study various properties of aerosol particles, fog droplets, and precursor gases, in order to better understand aerosol-fog interactions. This study focuses on the period of February and March 2022, when the mobile aerosol-cloud laboratory, a measurement container equipped with instrumentation to study aerosol particles and cloud/fog droplets, was at SPC.

### 2.2 Measurement site

The SPC research station (44°39'15"N, 11°37'29"E) is a rural measurement site, located in the Po Valley, about 30 km northeast of Bologna, Italy. Standard particle- and gas-phase, as well as meteorological parameters are monitored continuously at the site, both by the Regional Agency for Prevention, Environment, and Energy of Emilia-Romagna (ARPAE-ER), and the Italian National Research Council – Institute of Atmospheric Sciences and Climate (CNR-ISAC). Meteorological data, such as temperature, RH, and wind speed were provided by CNR-ISAC. Since the 1980s, numerous field studies have been conducted at SPC, focusing on aerosol physical and chemical processes in fog (Noone et al., 1992; Facchini et al., 1999; Gilardoni et al., 2014), OA and its sources (Gilardoni et al., 2016; Paglione et al., 2020), gas phase chemistry (Cai et al., 2024), and new particle formation (Lampilahti et al., 2021).

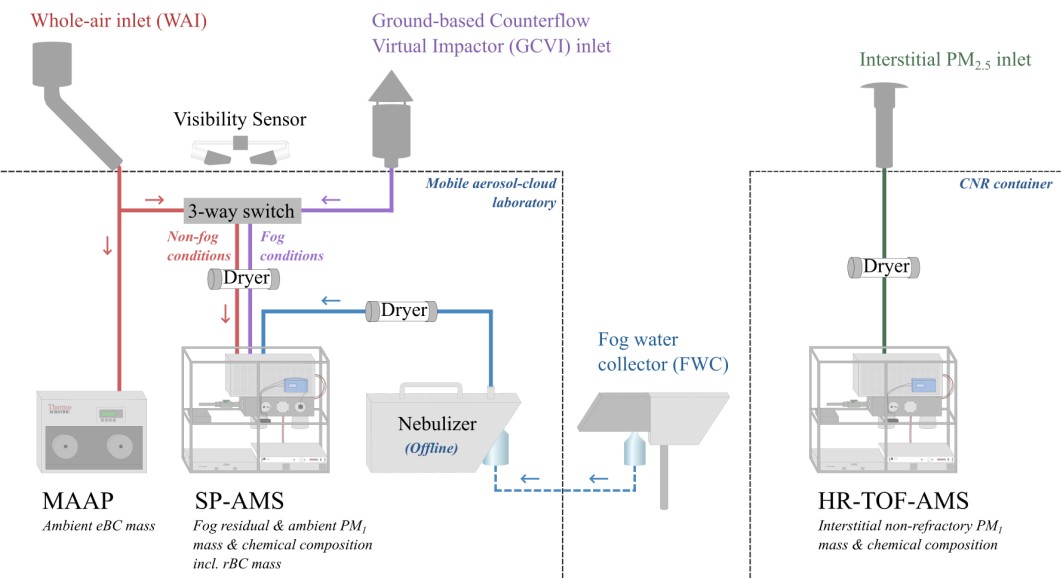

**Figure 1.** Schematic of the experimental set-up inside the two containers, including instruments relevant in this study. The two inlets in parallel, connected with a 3-way switch, allow selected instruments to change sampling inlet from the WAI to the GCVI during fog. The different colored lines represent how the instruments were connected during ambient aerosol sampling using the WAI (red), during fog events using GCVI inlet (purple), and the interstitial aerosol sampling (green). Offline fog water analysis was performed in the laboratory (blue).

## 2.3 The mobile aerosol-cloud laboratory

The mobile aerosol-cloud laboratory consists of a wide range of instruments for characterizing aerosol particles, fog, and the meteorological conditions. The indoor temperature was controlled and set to 21 °C. For more details about the instrumental setup inside the mobile aerosol-cloud laboratory, see Neuberger et al. (2025). The laboratory had two inlets sampling in parallel
(Fig. 1): a whole-air inlet (WAI) and a Ground-based Counterflow Virtual Impactor (GCVI; Brechtel Mfg. Inc., USA, Model 1205) inlet. The WAI sampled the total ambient aerosol with a flow of 99 lpm, of which 80 lpm was make-up flow. The inlet was heated to 20 °C until 2022-02-19, when it was changed to 30 °C.

The sampling of dried fog droplets (fog residuals) was done using the GCVI inlet. For a more technical description of the GCVI inlet, see Shingler et al. (2012) and Karlsson et al. (2021). In brief, the working principle of the GCVI is that two
opposing air flows (a sample flow and a dry, particle-free counterflow) separate particles depending on their inertia. First, the particles are accelerated onto the tip of the inlet by the wind tunnel. This concentrates the particles, resulting in an enrichment of the fog/cloud droplets compared to in the ambient air, which has to be corrected for. The interstitial aerosol is caught by an opposing counter-flow, while large enough particles, i.e. fog or cloud droplets, continue traveling through the GCVI, and are simultaneously dried. The resulting dried fog droplets are referred to as fog residuals in this study. During the FAIRARI
campaign, the lower cut size of the GCVI inlet was set to 7.7 μm aerodynamic diameter, while the upper size limit was

approximately 40 $\mu$m. The enrichment factor was 6.7, and the sample flow through the GCVI was kept at 15 lpm. It should be noted that we did not account for the droplet sampling efficiency of the GCVI, as suggested by Shingler et al. (2012) and Karlsson et al. (2021). Therefore, the sampled fog residual masses should be considered a lower limit of the real ambient values. Visibility was measured with a Visibility Sensor Model 6400 (Belfort Instrument, USA). Whenever the visibility was lower than 1 km, which follows the definition of fog by the World Meteorological Organization, the GCVI inlet was automatically activated, and simultaneously, a number of the instruments switched from measuring behind the WAI to behind the GCVI (Neuberger et al., 2025).

The total length of the sampling lines from the WAI and GCVI to the SP-AMS were 578 cm and 360 cm, respectively. The sampling line from the WAI consisted of 380 cm 1/4 inch (6.35 mm) inner diameter (ID) stainless steel tubing and 198 cm black conductive 1/4 inch ID tubing, while the GCVI sampling line consisted of 305 cm 1/4 inch ID stainless steel tubing and 55 cm 1/4 inch ID black conductive tubing. Behind the 3-way switch (Fig. 1), the sample flow was 2 lpm, where the SP-AMS flow was 0.1 lpm. The aerosol was consistently dried to below 30 % RH, using a 100 cm Nafion dryer model MD-700 (Perma Pure LLC, NJ, USA).

## 2.4  Aerosol chemical composition measurements

The Soot-Particle Aerosol Mass Spectrometer (SP-AMS; Aerodyne Research Inc., USA) (Onasch et al., 2012) is a High-Resolution Time of Flight Aerosol Mass Spectrometer (HR-TOF-AMS) (DeCarlo et al., 2006), equipped with a laser vaporizer in addition to the tungsten vaporizer. It measures the size-resolved aerosol mass concentration of the non-refractory submicron particulate matter (NR-PM$_1$), which includes OA, NO$_3^-$, SO$_4^{2-}$, NH$_4^+$, and Cl$^-$, as well as refractory black carbon rBC. Aerosol size distribution information is determined from the particles' time of flight using ePToF (efficient particle time-of-flight). Upon exiting the aerodynamic lens, the particles have a size-dependent velocity. The beam of particles hits a spinning chopper wheel with multiple slits. The time the particles need to go from an opening in the chopper to being acquired as several mass spectra in the mass spectrometer is used to determine the size of the particles (DeCarlo et al., 2006). Due to the additional laser vaporizer, based on the principle of the Single Particle Soot Photometer (SP2; Droplet Measurement Technology, USA, Stephens et al., 2003), the quantification of rBC is achievable with the same instrument.

Throughout the study, the SP-AMS was alternating between three sampling modes; a single-vaporizer V-mode (30 minutes), a dual-vaporizer V-mode (30 minutes), and a single-vaporizer W-mode (15 minutes). In the single-vaporizer V-mode, the tungsten vaporizer is heated to 600 °C in order to flash-vaporize the NR-PM$_1$. The vaporized particle components are ionized, using 70 eV electron impact ionization, and continue into the extraction region of the mass spectrometer. The ions are separated according to their mass to charge ratio (m/z), by following a V-shaped trajectory, first to the reflectron, and then back to the multichannel plate (MCP) detector. During dual-vaporizer mode, an intra-cavity Nd:YAG laser (1064 nm) is activated in addition to the tungsten vaporizer, which allows quantification of rBC particles. In the single-vaporizer W-mode, once again, only the tungsten vaporizer is activated, however, here, the ions entering the mass spectrometer follow a W-shaped trajectory. The ions are initially directed into a hard mirror, which makes the ions take a second flight to the reflectron before they reach

the MCP detector. This provides a higher mass resolution (m / $\Delta$m = 3500) compared to the V-mode (m / $\Delta$m = 2400) (DeCarlo et al., 2006).

All SP-AMS data were processed and analyzed using the AMS data analysis software packages SQUIRREL 1.66 and PIKA 1.26 (Sueper, 2024), together with Igor Pro 9 (Wavemetrics, USA). The data were recorded at a 1-minute resolution, which was used during analysis. The ionization efficiency was calculated from calibrations using 300 nm ammonium nitrate particles. For the rBC calibration, monodisperse rBC particles were created and nebulized using regal black mixed with deionized ultrapure Milli-Q water. The resulting relative ionization efficiencies for ammonium, nitrate, organics, sulfate, chloride, and rBC were 3.4, 1, 1.4, 1.2, 1.3, and 0.2, respectively. The collection efficiency was calculated using the composition-dependent collection efficiency approach (Middlebrook et al., 2012), and was on average 0.54 throughout the measurement campaign. Due to an underestimated ionization efficiency for the SP-AMS, the mass concentration measured by the SP-AMS had to be corrected (see Fig. S1). This was done using data from a collocated HR-TOF-AMS provided by CNR-ISAC (from now on called the interstitial AMS; see below for further details), which in turn was validated against offline ion-chromatography on parallel $PM_1$ filter samples. The underestimation of ionization efficiency likely originated from a faulty flow settings in the calibration setup of a Differential Mobility Particle Sizer (DMPS) used for the AMS calibration. The correction factor applied to all SP-AMS data was on average 0.51. All elemental ratios, such as oxygen-to-carbon (O:C) and nitrogen-to-carbon (N:C) were calculated using the improved ambient (IA) method within PIKA 1.26 (Canagaratna et al., 2015). In order to quantify the ON ion categories used in this study, $C_xH_yN_1^+$, $C_xH_yN_2^+$, and $C_xH_yO_1N_1^+$, the high resolution W-mode data were used during peak fitting. The W-mode was set to scan up to m/z 106, which means that only ON ions up to this m/z were included during the peak fitting.

In addition to the SP-AMS rBC measurements, aerosol light absorption was measured using a multi-angle absorption photometer (MAAP; Thermo Fisher Scientific, USA) (Petzold and Schönlinner, 2004), which was used to verify the rBC mass concentration from the SP-AMS. The mass absorption cross-section ($MAC_{BC}$) of $6.6\,\mathrm{m^2\,g^{-1}}$ was used to estimate the equivalent BC (eBC) mass concentration. During non-fog periods, both instruments measured behind the WAI, where they showed an overall strong linear correlation (Pearson correlation coefficient, r=0.91, see Fig. S2). The slope between the two instruments was 0.70, with the MAAP measuring on average higher mass concentrations than the SP-AMS. Since the MAAP measures at a single wavelength of 670 nm, there could be interference from e.g. brown carbon (BrC). In fact, a recent study by Shen et al. (2024) found that $23\,\%$ of total absorption at 660 nm was due to water-soluble BrC in an environment heavily influenced by biomass burning. Considering that the wintertime Po Valley aerosol is highly influenced by biomass burning emissions, there could be interference from BrC, to the eBC measured by MAAP.

Data from a second HR-ToF-AMS (interstitial AMS) were used to provide complementary information on the interstitial aerosol (i.e., non-activated aerosol particles) chemical composition during fog events. The data from the interstitial AMS were analysed following the same protocol as described above for the SP-AMS, but the instrument was installed in a different container and sampled aerosol particles through a $PM_{2.5}$ inlet. Therefore, during fog events, this AMS sampled only the interstitial (unactivated) particles, while fog droplets (activated particles) could not reach the sampling line because of the smaller cut-off of the sampling head (Neuberger et al., 2025). In addition, the aerosol liquid water content (ALWC) was

predicted using the aerosol inorganic species; $NO_3^-$, $SO_4^{2-}$, $NH_4^+$, and $Cl^-$ measured by HR-ToF-AMS, as well as $K^+$, $Ca^{2+}$, $Mg^{2+}$, and $Na^+$ from offline ion-chromatography on $PM_1$ filter samples, as inputs in ISORROPIA-II model (Fountoukis and Nenes, 2007).

## 2.5 Offline analysis of fog water and $PM_1$ samples

Fog water was sampled at the SPC field station using a Fog Water Collector (FWC), described in more detail by Fuzzi et al. (1997), and previously used in Paglione et al. (2021). Briefly, the FWC is an automated active string collector and sampling is dictated automatically by a built in detector. Each individual string has a $50\%$ collection efficiency at approximately $3\,\mu m$ droplet radius. The threshold value for fog has previously been validated with liquid water content measurements (Fuzzi et al., 1997). The fog droplets enter the collector via an airflow of about $17\,m^3\,min^{-1}$, created by a fan placed in the back of the FWC. The fog droplets impact on the stainless-steel strings, leading the droplets into a glass bottle via a funnel. After collection, the fog water samples were filtered through $47\,mm$ quartz-fiber filters and kept frozen until they were analyzed.

The offline analysis of the fog water samples took place in the laboratory, where each sample was nebulized using a Topas ATM 228 Aerosol Generator (Topas GmbH, Dresden, Germany) and measured with a SP-AMS. The aerosol was dried with a diffusion dryer, and the RH was kept below $20\%$ throughout the experiments. In total, 9 fog water samples were collected in February and March 2022. Each fog water sample was analyzed using this setup for at least one hour, while the SP-AMS altered between the same three sampling modes as during the ambient and fog residual measurements. In order to be sure of no contamination from either previous samples or the background, blank samples of deionized ultrapure Milli-Q water were measured for at least 10 minutes between each fog water sample. The measured signal from the blank samples was negligible. During the analysis of the fog water samples, relatively large peaks were observed in the mass spectra at m/z 73, 143, 207, 221, and 281. These peaks are likely artifacts from the conductive silicone tubing (Timko et al., 2009), and thus excluded from the analysis.

Sub-micron particles were also sampled on pre-washed and pre-baked quartz-fiber filters (PALL, 15 cm size) using a HiVol sampler (TECORA, Italy), equipped with a $PM_1$ sampling head (Digitel Elektronik AG, Switzerland), using a nominal flow of $500\,l\,min^{-1}$, located at ground level (Neuberger et al., 2025). The HiVol quartz-fiber samples were analyzed to chemically characterize the water soluble organic carbon and to identify 1H-imidazole, for the first time at SPC, using proton nuclear magnetic resonance ($^1$H-NMR) spectroscopy following the method described by Decesari et al. (2024). This method was also used to investigate the presence of 1H-imidazole in the fog water samples. Briefly, the sampled aerosol was extracted with deionized ultrapure Milli-Q water in a mechanical shaker for 1 hour and aliquots of these extracts and of fog waters were dried under vacuum and re-dissolved in deuterium oxide ($D_2O$) for the analysis. The $^1$H-NMR spectra were acquired at $600\,MHz$ in a $5\,mm$ probe using a Varian Unity INOVA spectrometer, at the NMR facility of the Department of Industrial Chemistry (University of Bologna). $^1$H-NMR spectroscopy in protic solvents provides the speciation of hydrogen atoms bonded to carbon atoms. On the basis of their frequency shifts, the signals can be attributed to H-C containing specific functionalities (Decesari et al., 2000, 2007) and/or to specific molecules (Suzuki et al., 2001; Paglione et al., 2024). The addition of an internal standard of known concentration allows the quantification of these specific molecular tracers when well-resolved individual resonances

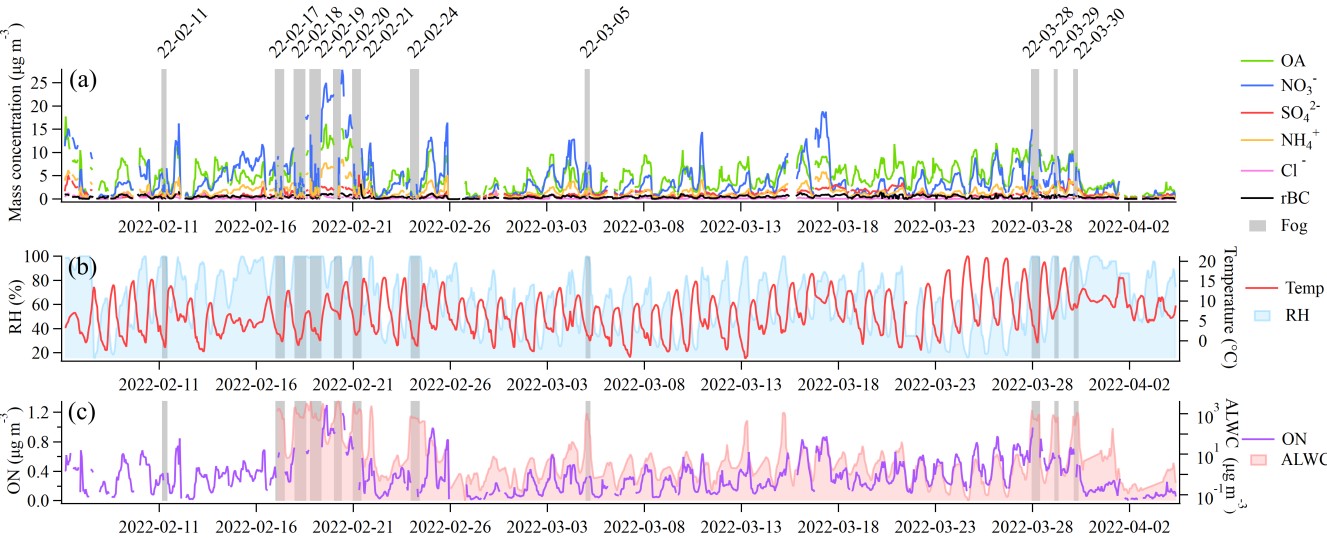

**Figure 2.** Overview of main parameters measured during the FAIRARI campaign from February to March, 2022. (a) shows the mass concentrations of aerosol chemical species recorded by the SP-AMS. The fog events are highlighted in gray shading, in which the SP-AMS measured the fog residuals behind the GCVI inlet. (b) shows the hourly mean temperature and relative humidity (RH). (c) shows the hourly averaged mass concentrations of the ON and the calculated ALWC.

match the spectra of standard compounds, such as 1H-imidazole, characterized by specific singlets at known chemical shifts. Sodium 3-trimethylsilyl- (2,2,3,3-d$_4$) propionate (TSP-d$_4$) was used as an internal standard by adding 50 $\mu$L of a 0.05 % TSP-d$_4$ (by weight) in D$_2$O to the standard in the probe. To avoid the shifting of pH-sensitive signals, the extracts were buffered to pH 3 using a deuterated-formate/formic-acid (DCOO$^-$=HCOOH) buffer prior to the analysis.

## 3 Results

### 3.1 Time series of aerosol species, meteorological parameters, and fog events

During February and March 2022 of the FAIRARI campaign, a total of 11 fog events were measured with the aforementioned setup (see Sect. 2.3 for the definition of a fog event and Table S1 for the exact times included in each fog event). Figure 2 presents an overview of the measurements, where Fig. 2a shows the aerosol mass concentration of the different aerosol species measured with the SP-AMS, with the fog events marked as gray bars. Over the two months, the average ambient total PM$_1$ mass concentration was 13.5 $\pm$ 9.1 $\mu$g m$^{-3}$, which consisted of: OA = 5.3 $\pm$ 3.0 $\mu$g m$^{-3}$, NO$_3^-$ = 4.4 $\pm$ 4.3 $\mu$g m$^{-3}$, SO$_4^{2-}$ = 1.1 $\pm$ 0.8 $\mu$g m$^{-3}$, NH$_4^+$ = 1.6 $\pm$ 1.4 $\mu$g m$^{-3}$, Cl$^-$ = 0.2 $\pm$ 0.2 $\mu$g m$^{-3}$, and rBC = 0.5 $\pm$ 0.3 $\mu$g m$^{-3}$. The total PM$_1$ mass concentration and the fractional composition were generally within the range of previous studies from the Po Valley (Paglione et al., 2020; Scotto et al., 2021). Compared to previous winter-time measurements performed specifically at SPC, the average PM$_1$ mass concentration and the contribution of OA (39.4 %) were on average lower, while the contributions from the inorganic

species: $NO_3^- = 34.6\%$, $NH_4^+ = 11.7\%$, and $SO_4^{2-} = 6.6\%$ were slightly elevated in the current study (Carbone et al., 2010; Gilardoni et al., 2014).

During fog, the SP-AMS automatically switched from the WAI to the GCVI in order to measure the fog residuals, which on average consisted of: $OA = 27.6\%$, $NO_3^- = 43.3\%$, $NH_4^+ = 15.2\%$, $SO_4^{2-} = 10.5\%$, and rBC = 2.3 %. During the same fog events, the average interstitial aerosol consisted of $OA = 32.9\%$, $NO_3^- = 37.1\%$, $NH_4^+ = 14.1\%$, $SO_4^{2-} = 4.6\%$, and BC = 10.6 %. The mass scavenging efficiency ($\eta$, Eq. 1) was calculated using the average mass concentration of each chemical species before fog formation and the interstitial aerosol (Noone et al., 1992).

$$\eta = 1 - \frac{[X]_{\text{interstitial}}}{[X]_{\text{before fog}}} \tag{1}$$

OA and BC were associated with the lowest mass scavenging efficiencies, on average 49 and 36 %, respectively. $NO_3^-$, $NH_4^+$, and $SO_4^{2-}$ were more readily scavenged by the fog, with average $\eta$ = 57, 59, and 57 %, respectively. Relative to both interstitial and ambient aerosol, a higher contribution of ammonium nitrate and ammonium sulfate were observed in the fog residuals, while OA and rBC represented a lower fraction. These results align with what was expected; partly that the already existing,
more hygroscopic, inorganic aerosol was preferred as CCN during the formation of the fog, and partly that ammonium nitrate and ammonium sulfate are more readily taken up by the fog droplets.

Figure 2b shows the variations in the hourly mean temperature and RH. During the first intensive fog period (February 16-22), the temperature was on average 8.8 °C and during the fog events 2.8 °C. In the second intensive fog period (March 27-30), the average temperature was 11.7 °C, whereas during the fog events it was 5.3 °C. The first intensive fog period was
characterized by a larger aerosol mass concentration, especially inorganic species, compared to the second period. This aerosol mass consisted of more ON and a higher aerosol liquid water content (ALWC) (Fig. 2c). By performing a Spearman's rank correlation test, a moderate correlation was found between the non-fog ambient aerosol ON mass concentration and the ALWC ($r_s$ = 0.44, p = 0.052), consistent with previous studies (Liu et al., 2023). This moderate correlation indicates that the ions included in this ON family, likely consisted of water-soluble ON compounds.

## 3.2   Comparison of two pollution episodes intersected by fog

Figure 3 shows the chemical composition of the aerosol at different stages of two periods with reoccurring fog events (February 16-22 and March 27-30), investigated in more detail here. Of these two periods, the first had particularly high $PM_1$ loadings as compared to the second one - offering a good case study for comparing aerosol-fog interactions with varying levels of pollution. The first, more polluted, intensive fog period (February 16-22) was initiated by a stagnation of the atmosphere (Neuberger
et al., 2025), and consequent build-up of pollution starting on Feburary 17 (Fig. 3a). This build-up persisted until February 20, independently of the fog events occurring every night. The $PM_1$ mass consisted of mainly $NO_3^-$, OA, and $NH_4^+$, while rBC and $SO_4^{2-}$ stayed constant over this period. Starting from February 20, the wind intensified, resulting in a dissipation of the pollution. Both the ambient and the fog residual $PM_1$ mass were dominated by inorganic species, especially $NO_3^-$ (Fig. 3b). The fraction of OA and rBC decreased in the fog residuals, while the $NH_4^+$, $SO_4^{2-}$ and $NO_3^-$ showed similar or slightly higher
mass fractions in fog residuals compared to ambient.

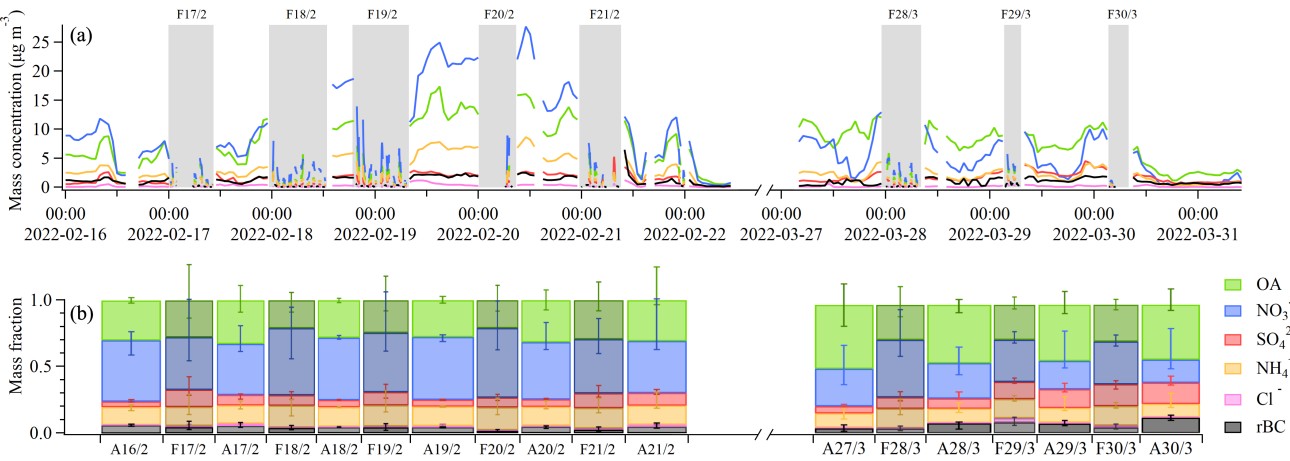

**Figure 3.** Intensive fog episodes between February 16-22 and March 27-30. (a) shows the aerosol mass time series from the SP-AMS. Fog events are indicated by gray bars. The ambient out-of-fog data are averaged over 1 hour, while the in-fog residual data are averaged over 5 minutes. (b) shows the median mass fraction for each day, with the ambient (A) and fog residuals (F) separated. The width of each bar corresponds to the duration in (a), and the error bars represent the 75[th] and 25[th] percentile.

In the second intensive fog episode (March 27-30), the ambient $NO_3^-$ fraction was comparatively smaller than in the first episode, while the ambient fraction of OA, $SO_4^{2-}$, and rBC was larger than in the first episode. The ambient fraction of $NH_4^+$ was similar in both episodes. The higher ambient ratio of OA to $NO_3^-$ in the second pollution episode is however not reflected in the fog residual composition; this looks very similar to the composition of the residuals in the first pollution episode with relatively higher nitrate contributions. During both episodes, the fog residuals were relatively enriched with inorganic components as compared to the ambient aerosols.

### 3.3  Chemical composition of ambient aerosol, interstitial aerosol, fog residuals, and fog water

We now compare in more detail the chemical composition of the average ambient aerosol before fog and after fog, and the interstitial aerosol to the fog residual and fog water composition (Fig. 4a). The ambient aerosol represents the average aerosol over the measurement campaign, excluding fog periods. Before and after fog is defined here as 1 - 2 hours before or after a fog event, while the interstitial (non-activated) aerosol is measured during the same fog events, by the interstitial AMS. Figure 4a reveals a scavenging of the inorganic species $NO_3^-$, $SO_4^{2-}$, and $NH_4^+$ from the interstitial mass fraction during fog. Simultaneously, these species are found enhanced in the fog residuals (68 % of the $PM_1$ mass in fog), while most of rBC remained non-activated. The fractional compositions of the major chemical species before and after fog are relatively similar, showing that most of the particles taken up by the fog return to the particle phase upon fog dissipation. Furthermore, before and after fog showed a higher contribution of $NO_3^-$ compared to the average ambient non-fog periods, while fog water showed the highest fraction of total organic carbon.

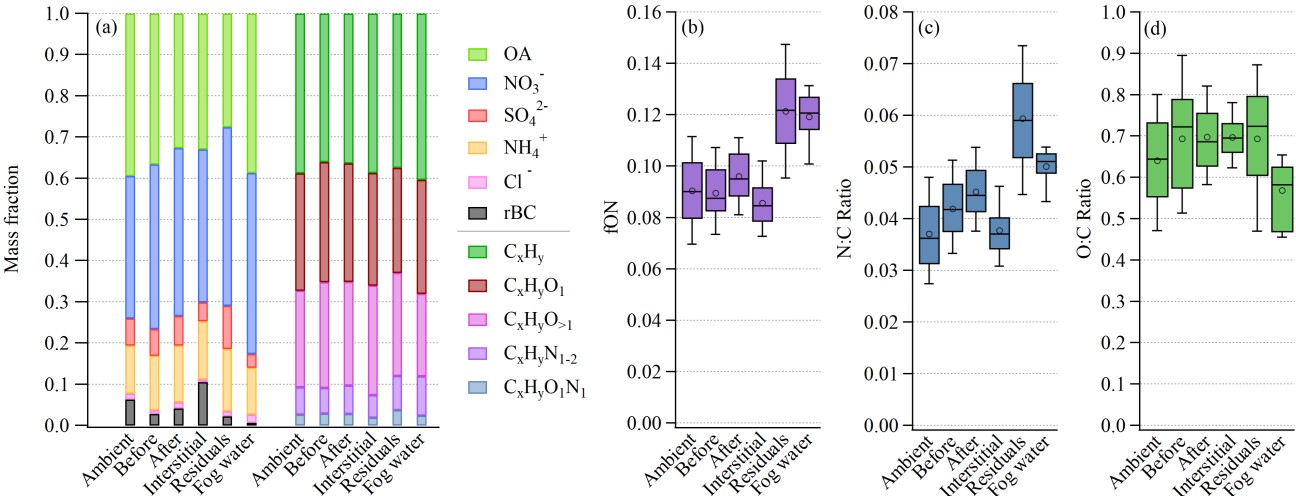

**Figure 4.** (a) Average mass fraction of the major aerosol species (left side) and the OA families (right side), box plots showing (b) the organic fraction of ON, (c) the N:C ratio, and (d) the O:C ratio. For all panels, the data are divided into 6 categories: ambient aerosol, before fog, after fog, interstitial, fog residuals, and fog water. For panels (b), (c), and (d), the whiskers above and below the boxes indicate the 90th and 10th percentile, the boxes indicate the 75th and 25th percentiles, the circles indicate the mean, and the lines indicate the median.

Further investigating the organic mass fraction (Fig. 4a), according to its OA families (fragment elemental composition groups), a similar composition was observed in the ambient, before, after, interstitial, and fog residuals, except for the nitrogen-containing OA families ($C_xH_yN_{1-2}^+$ and $C_xH_yO_1N_1^+$), which were enhanced in the fog residuals. This is also shown in Fig. 4b, where the fON is the organic fraction of ON ions. The fog residuals, measured behind the GCVI, show a distinct enhancement of ON (Fig. 4b), compared to the ambient aerosol. The nitrogen-to-carbon (N:C) ratio (Fig. 4c) was significantly higher in the fog residuals, highlighting the elevated organic nitrogen present in the fog residuals compared to the ambient aerosol.

Similar to the fog residuals, the nebulized fog water samples showed an enhanced nitrogen-containing OA ($C_xH_yN_{1-2}^+$ and $C_xH_yO_1N_1^+$). Interestingly, this did not result in an equally high N:C ratio in the fog water, despite the high fraction of ON, indicating higher contributions of compounds with higher carbon numbers in the fog water compared to the residuals. This discrepancy is also reflected in the lower oxygen-to-carbon (O:C) ratio of the fog water, shown in Fig. 4d - in line with the organic mass fractions in Fig. 4a. In fact, the fog water samples contained a larger fraction of hydrocarbons ($C_xH_y^+$), mainly m/z 27 and 41, compared to the other categories. Simultaneously, the fraction of more oxidized OA ($C_xH_yO_{>1}^+$) was smaller compared to the other categories, resulting from a lower m/z 44 ($CO_2^+$) signal in the fog water compared to the fog residuals (Fig. S3). This could be, at least partly, due to the GCVI and WAI sampling inlets, where more volatile aerosol components could evaporate and partition into the gas phase due to the heating and drying, resulting in a higher O:C ratio for the fog residuals compared to fog water. However, the Po Valley fog water was expected to be relatively more oxidized. The fog water O:C ratio was previously estimated to $0.68 \pm 0.08$ at SPC (Gilardoni et al., 2016), which is similar to the current study's O:C ratio in the ambient, interstitial, and fog residuals. A previously observed O:C ratio in fog water from Fresno, California,

was $0.42 \pm 0.04$, which is less oxidized than the Po Valley fog, but more similar to the Fresno pre- and post-fog (Kim et al., 2019). At a polluted site in Kanpur, India, the fog water O:C ratio was estimated to $0.68 - 0.88$, which again is more in line with the current study's O:C ratio for the other categories (Chakraborty et al., 2016). Another hypothesis is that the chemical composition was not consistent over all droplet sizes, which could explain the disparity between the fog water and residuals, due to the different droplet size range of the two sampling methods, (i.e. FWC and GCVI). Further research is needed on the discrepancies that arise from different fog/cloud measurement techniques.

## 3.4 Characterization of enhanced ON in fog residuals and fog water

The ON ions were divided into three OA families; $C_xH_yN_1^+$, $C_xH_yN_2^+$, and $C_xH_yO_1N_1^+$. The ON mass spectra observed in the fog water (Fig. 5b) and ambient aerosol (Fig. 5c) show similarities to ON mass spectra observed in wintertime Fresno, California (Kim et al., 2019; Ge et al., 2024). Both of these studies utilized an HR-TOF-AMS together with an atomizer for the offline analysis of fog water samples, similar to the fog water analysis in this study. The $C_xH_yN_1^+$ ions dominate all four ON mass spectra (Fig. 5), where a few of the observed peaks (e.g. $CH_4N^+$ and $C_2H_6N^+$) are tracer fragments of amines (Ge et al., 2024). Most of the ON ions that appear in the fog residuals and fog water show similar relative abundance (Fig. S4), especially the lower m/z range ions with the highest signals (e.g. $C_1H_1N_1^+$ and $C_1H_1N_1O_1^+$). Considering all ON ions identified in this study, a strong correlation was found between the fog residuals and the fog water mass spectra (Pearson correlation coefficient, r=0.89). While most $C_xH_yO_1N_1^+$ ions constitute a larger fraction in the fog residuals, the $C_xH_yN_2^+$ ions show an opposite tendency.

$C_xH_yN_1^+$ ions were previously found as a major constituent in organic aerosol via source apportionment (Saarikoski et al., 2012; Sun et al., 2011). During a spring-time field campaign at SPC in the Po Valley, Saarikoski et al. (2012) found a nitrogen-enriched OA (NOA) factor, where $14\,\%$ of the factor was comprised of $C_xH_yN_1^+$ ions. This factor had a N:C ratio of 0.078, slightly higher than the fraction in the fog residuals in our study (Fig. 4c). They argue that this factor came from local traffic sources due to its diurnal trend and sharp decrease during the nocturnal surface layer break-up. This diurnal trend is neither observed in the N:C ratio nor for any ON compounds in this study. During non-fog conditions, the N:C ratio is in general high during the day and low during the night, reaching its maximum around 14:00 local time. During the first fog and pollution week, the N:C ratio shows less of a diurnal pattern, and stays at its maximum throughout the day.

A smaller fraction of the ON consisted of $C_xH_yN_2^+$ ions, ranging from $5\,\%$ (ambient) to $10\,\%$ (fog water). From the NIST (National Institute of Standards and Technology Standard Reference) database on electron impact ionization mass spectra it can be found that certain imidazoles have characteristic $C_xH_yN_2^+$ fragmentation peaks. The three $C_xH_yN_2^+$ peaks giving the largest signal in the fog water mass spectra match with major signals for different imidazoles. Specifically, the $C_3H_4N_2^+$ peak at m/z 68 is likely 1H-imidazole, while the $C_4H_6N_2^+$ peak at m/z 82 is likely related to 2-Methyl-1H-imidazole, and $C_5H_7N_2^+$ at m/z 95 could be a fragment of 3,5-Dimethylpyrazole.

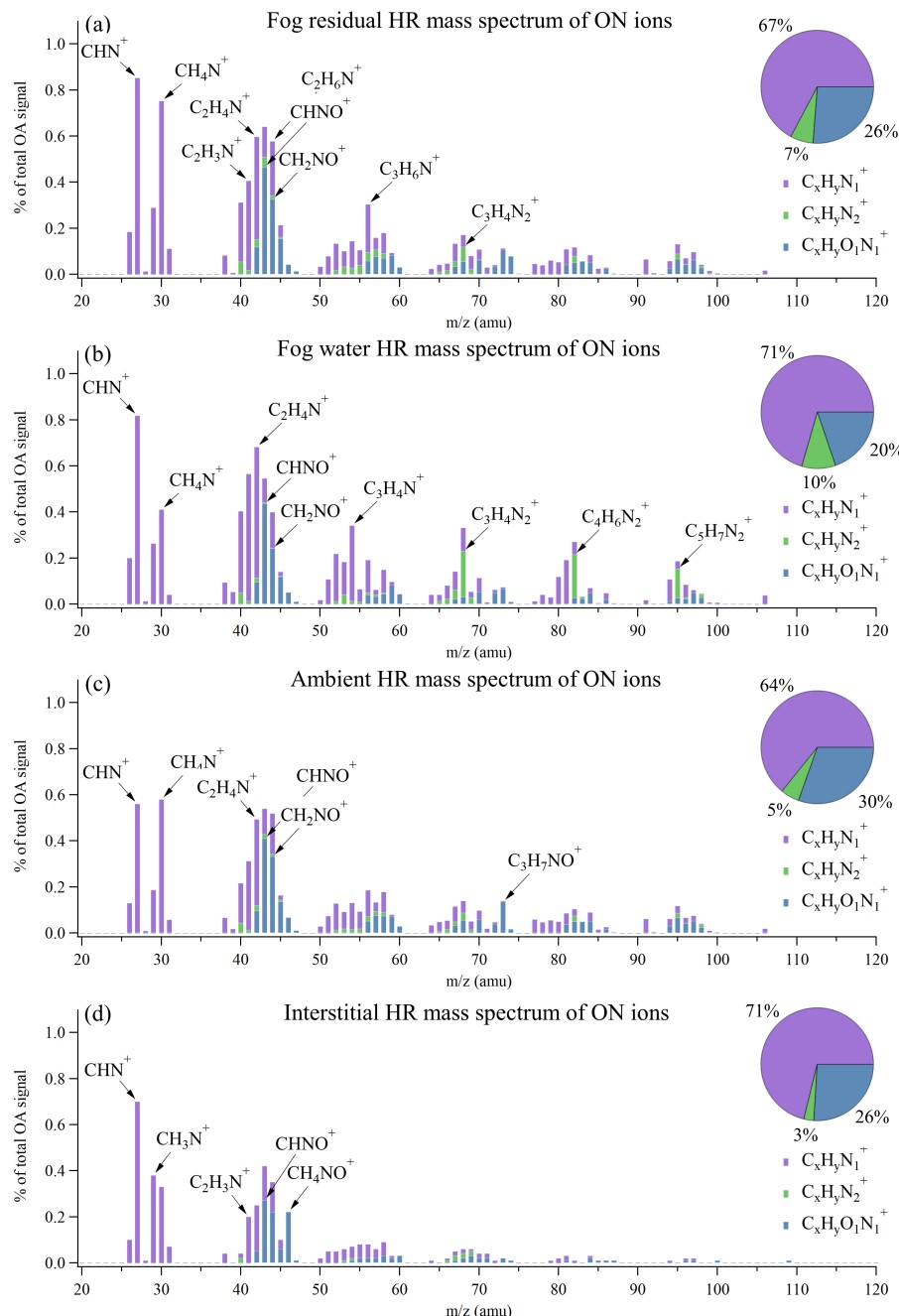

**Figure 5.** Average high resolution mass spectra of the nitrogen-containing organic ions presented as percentage of total OA signal in (a) fog residuals, (b) fog water, (c) ambient aerosol, and (d) interstitial aerosol. The pie charts show the fraction of each category ($C_xH_yN_1^+$, $C_xH_yN_2^+$, and $C_xH_yO_1N_1^+$).

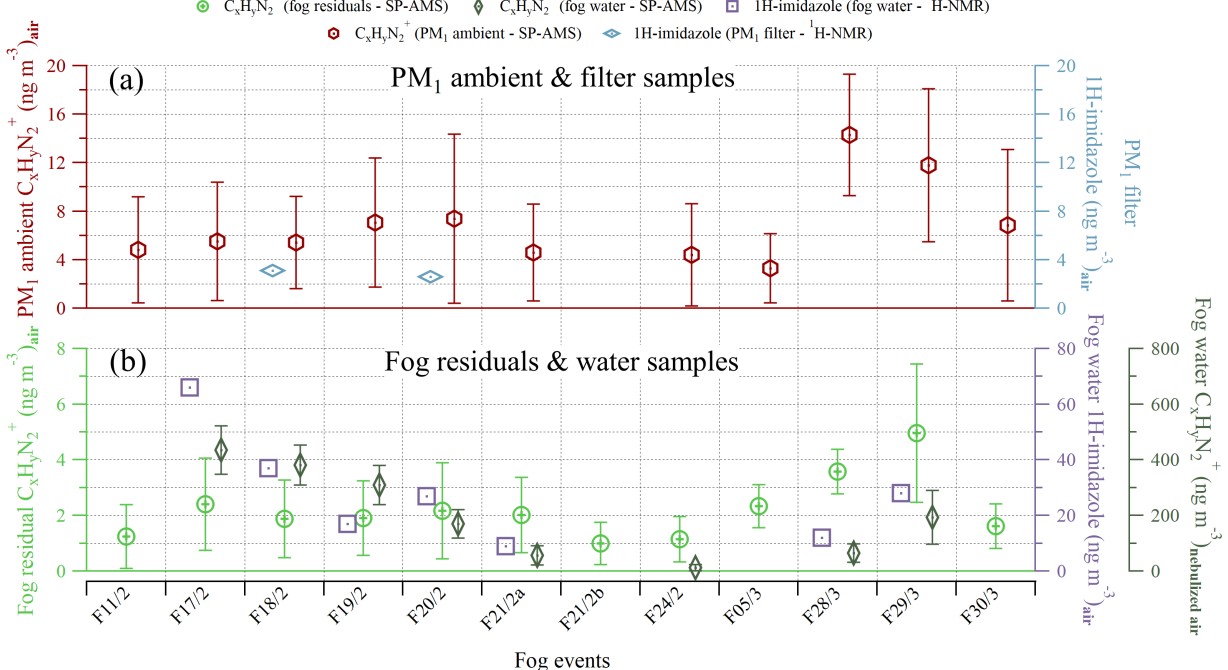

**Figure 6.** Mass concentration of $C_xH_yN_2^+$ measured by SP-AMS, and 1H-imidazole by [1]H-NMR spectroscopy for each fog event. Panel (a) shows the sub-micron ambient $C_xH_yN_2^+$ (red, left y-axis) and the 1H-imidazole from offline filter samples (light blue, right y-axis). Panel (b) includes the fog residual $C_xH_yN_2^+$ (light green, left y-axis), fog water $C_xH_yN_2^+$ (dark green, right y-axis), and fog water 1H-imidazole (purple, right y-axis). The fog water 1H-imidazole concentration in mass per m$^3$ of air was obtained by multiplying its concentration in fog water with the liquid water content of the air, measured by a Particulate Volume Monitor. The fog water $C_xH_yN_2^+$ concentration is displayed in mass per m$^3$ of nebulized air, and is thus only used to look at the trend. Each symbol represents the mean, while the error bars represent one standard deviation.

The presence and concentration of 1H-imidazole were confirmed through [1]H-NMR analysis conducted on the PM$_1$ filter samples and fog water samples (Fig. 6). This compound produced two distinct singlets at 7.47 and 8.69 ppm of chemical shift and was identified by comparison with a 1H-imidazole standard under identical [1]H-NMR experimental conditions (Fig. S5).
The absolute concentration of 1H-imidazole should be considered a tentative quantification, due to possible re-evaporation not taken into account yet. However, the trends may be considered reliable. Notably, 1H-imidazole was found in all fog water samples at relatively high levels, while only trace amounts were detected in the offline PM$_1$ samples using the [1]H-NMR analysis. This strongly suggests that the compound formed under aqueous conditions, possibly through the dissolution of the gaseous fraction into fog droplets or water. Additionally, the concentrations obtained via [1]H-NMR show reasonable similarities
with those measured by SP-AMS, both in the ambient aerosol (Fig. 6a) and in the fog (Fig. 6b). The fog water $C_xH_yN_2^+$ and fog water 1H-imidazole exhibit especially similar trends. Imidazole formation from aqSOA reactions is known to occur in the atmosphere based on laboratory studies (Herrmann et al., 2015). The first observation of imidazole in laboratory-generated

aerosol particles was done by Laskin et al. (2009), while the first in-situ observation of imidazole in the particle phase was reported by Teich et al. (2016). Recent experiments focusing on the characterization of ON in fog water have suggested the attribution of AMS fragments to imidazoles (Kim et al., 2019; Ge et al., 2024), which were also found in the current study. However, the formation of imidazole by aerosol-fog interactions has previously never been confirmed by a molecular-level analysis integrated with online observation using aerosol mass spectrometry.

Previous laboratory studies have found evidence of aqueous-phase production of ON compounds (such as imidazoles) from uptake of glyoxal with amino acids (De Haan et al., 2009) or ammonium (Galloway et al., 2009). Glyoxal, an important precursor for aqSOA formation, typically fragments into $C_2H_2O_2^+$, $C_2H_2^+$, $CH_2O_2^+$ from electron impact ionization (Paglione et al., 2020). Over the full measurement period, no significant linear correlation (Pearson correlation coefficient) was found between the $C_xH_yN_2^+$ ions and the glyoxal fragments in the ambient aerosol (r=0.38) and in the fog residuals (r=0.37). Similarly, the $C_xH_yN_2^+$ ions did not show a correlation with $NH_4^+$ in the ambient aerosol (r=0.26) and in the fog residuals (r=0.29). However, considering only the second intensive fog period (March 27-30), the $C_xH_yN_2^+$ ions were much more abundant in the ambient aerosol and slightly more abundant in the fog residuals. During this period, stronger correlations were found in both the ambient aerosol (r=0.65 and r=0.64 with glyoxal and $NH_4^+$, respectively) and in the fog residuals (r=0.70 and r=0.67 with glyoxal and $NH_4^+$, respectively). Throughout the campaign, the concentration of gas-phase $NH_3$ was on relatively similar levels, fluctuating between 10 and 30 ppb (see Fig. S6). No considerable difference was found between the first (17.3 ppb) and second (15.9 ppb) intensive fog period for $NH_3$. While data were missing for more than half of the first period, it was found reasonable to assume that the $NH_3$ should follow a similar trend over those days. Furthermore, the high concentration of $NO_3^-$ during the first intensive fog period suggests that $HNO_3$ was a large sink for $NH_3$. During the second intensive fog period, however, the $NO_3^-$ concentration was considerably lower, which could mean more $NH_3$ / $NH_4^+$ was available for imidazole formation. Together with the more OA-dominated aerosol, this could have enhanced the formation of imidazoles through glyoxal uptake with ammonium, suggested both by the increased abundance of imidazole fragments and the stronger correlations with glyoxal and $NH_4^+$ during the second intensive fog period. Meanwhile, good linear correlations were found between the $C_xH_yN_1^+$ ions and glyoxal fragments throughout the campaign (ambient r=0.80 and fog residuals r=0.84). Since glyoxal is a precursor of aqSOA, this suggests that a considerable part of the ON compounds are formed, or partition into the aerosol, during similar conditions as for aqSOA formation.

For imidazoles, little is known about their physicochemical properties, such as volatility. According to UK REACH (2018), 1H-imidazole has a vapor pressure of 3.27 x $10^{-10}$ Pa, which would mean that it has a low volatility. A more recent study estimated imidazoles in general to have a vapor pressure of 0.2 Pa or more, which instead would mean that they are highly volatile (Amugoda and Davies, 2025). However, they show that imidazoles stay in the particle phase, despite the high volatility, when they are internally mixed with inorganic salts. Therefore, imidazoles are expected to remain in the particle phase after droplet evaporation in the different inlets and drying of the aerosol. De Haan et al. (2011) found formation of imidazoles, from nitrogen-containing species (i.e. ammonium and amines) and glyoxal, during simulated evaporation of droplets. Ervens et al. (2011) summarized that these reactions likely occur in the evaporating droplets, when glyoxal has dissolved in the water and, simultaneously, the evaporation of water results in increasing concentrations of the nitrogen-containing compounds. A possible

explanation why the $C_xH_yN_2^+$ ions were reduced in the online fog residual measurements, compared to fog water samples, could be that they were more concentrated in the smaller evaporating fog droplets, which were sampled more efficiently by the FWC compared to the GCVI.

## 4 Conclusions

Aqueous-phase processing of aerosol particles and fog chemistry was studied in the Po Valley during winter-time, as part of the FAIRARI 2021/22 campaign. The sub-micron aerosol chemical characterization was carried out using one SP-AMS, for the ambient aerosol and fog residuals, and one HR-TOF-AMS, for the interstitial aerosol. The ambient $PM_1$ concentration was on average 13.5 ($\pm$ 9.1) $\mu g\,m^{-3}$ throughout the campaign, which included 39 % OA, 33 % $NO_3^-$, 8 % $SO_4^{2-}$, 12 % $NH_4^+$, 1 % $Cl^-$, and 4 % rBC. Compared to previous winter-time measurements at the site, the $PM_1$ mass was slightly lower, while still being dominated by OA, although to a lesser degree.

During two pollution episodes, consisting of high relative humidity and high levels of pollution, several fog events were captured. The sampling of the dried fog droplets (fog residuals) was done by applying a ground-based GCVI inlet. While the chemical composition of the fog residuals was relatively similar over these periods, the ambient aerosol was $NO_3^-$-dominated in the first period, whereas OA dominated the ambient aerosol in the second period. This was likely due to the on average lower temperature and simultaneously higher RH during the first period, which allowed more gas-particle partitioning of especially inorganic compounds. On average over the measurement campaign, $NO_3^-$ showed slightly larger variation due to temperature, indicating that it is more semi-volatile than the OA.

Fog water samples were collected and analyzed with an offline nebulizer setup, using the same SP-AMS. The fog water contained an unexpectedly large fraction of total OA, especially hydrocarbons, while the signal of more oxygenated OA was lower. This stood out in contrast to the ambient aerosol and fog residuals, and resulted in lower than expected N:C and O:C ratios, not consistent with previous fog water studies. Similar to this study, previous fog/cloud studies found that the mass scavenging efficiency is lower for OA than e.g. $NO_3^-$ and $SO_4^{2-}$, and that the cloud residual mass fraction of OA typically is lower than that of the ambient aerosol. Furthermore, the mass concentration of the OA families measured by the interstitial AMS and the SP-AMS agreed well during the non-fog periods. Thus, we hypothesize this discrepancy could be due to the different sampling methods and that the chemical composition could be dissimilar in larger versus smaller fog droplets.

A substantial enhancement of ON was found in both fog residuals and fog water samples, which also proved to be highly correlated. The fog-enhanced ON consisted of several $C_xH_yN_1^+$ ions, likely originating from various amino compounds, while a few ions could be identified as primary and secondary amines. These ON compounds are likely products from biomass burning, agriculture, and animal husbandry emissions. The increased abundance of these compounds in the fog, while also showing good correlation with glyoxal fragments and moderate correlation with ALWC, suggests that a considerable amount of the ON observed in the particle phase are either produced by aqueous-phase chemical reactions, or partitioning between the gas and aqueous phase.

The $C_xH_yN_2^+$ ions, mainly present in the fog water, suggest formation of imidazoles in the fog droplets. The related fragments linked to imidazoles have been confirmed by [1]H-NMR spectroscopy, and thus can now be used as tracer fragments. We hypothesize the reason they were less present in the fog residuals, compared to the fog water, could be that they were more concentrated in the smaller evaporating fog droplets, sampled more efficiently by the FWC compared to the GCVI. In the second intensive fog period, these $C_xH_yN_2^+$ ions were substantially elevated in the fog residuals, and even persisted in the aerosol phase after fog dissipation, which was not observed during the first intensive fog period. During this period, the $C_xH_yN_2^+$ ions showed strong positive correlation with both glyoxal fragments and $NH_4^+$, which was not found during other periods. Compared to the first intensive fog period, the aerosol mass consisted of a lower $NO_3^-$ concentration, while the fraction of OA was considerably higher, and $NH_4^+$ stayed relatively constant. This could explain the elevated mass concentration of $C_xH_yN_2^+$ ions, since more $NH_3$ / $NH_4^+$ was available for aqueous reactions with glyoxal, or other $\alpha$-dicarbonyls, to form imidazoles. Since the formation of 1H-imidazole is irreversible, even small amounts could have an effect on the aerosol optical properties. This shows the importance of fog processing from both an air quality and climate perspective. The results from this study improve our understanding on fog processing, and demonstrate that fogs and clouds are an important medium for aqueous production of certain nitrogen-containing organic aerosol in the atmosphere.

*Data availability.* All data, including HR-AMS mass spectra, will collectively be available at the Bolin Centre for Climate Research data base.

*Author contributions.* FM, PZ, IR, and CM designed the study. FM, AN, LH, YG, MP, MR, SD, PZ and CM collected the data at the SPC site. FM processed and analyzed the SP-AMS data. AN and PZ processed the GCVI data. MP processed and analyzed the [1]H-NMR spectroscopy data. MR processed and analyzed the interstitial HR-TOF-AMS data. FM performed the overall analysis, data visualization, and wrote the paper with input from CM, IR, and PZ. All authors gave comments and suggestions on the paper.

*Competing interests.* Some authors are members of the editorial board of journal Atmospheric Chemistry and Physics.

*Acknowledgements.* Financial support from the European Union's Horizon 2020 research and innovation programme (project FORCeS under grant agreement No 821205, "NPF-PANDA" under grant agreement No. 895875), European Research Council (Consolidator grant INTE-GRATE No 865799), Knut and Alice Wallenberg Foundation (grant numbers 2021.0169 and 2021.0298) are gratefully acknowledged. We thank Maurizio Busetto and David Hadden for their technical support in the field.

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
