# Peer review of "Enrichment of organic nitrogen in fog residuals observed in the Italian Po Valley"

_EGUsphere, 2024_

## Author Comment (AC1)

**Enrichment of organic nitrogen in fog residuals observed in the Italian Po Valley - Review comments**

Mattsson, F., Neuberger, A., Heikkinen, L., Gramlich, Y., Paglione, M., Rinaldi, M., Decesari, S., Zieger, P., Riipinen, I., and Mohr, C. (2024). Enrichment of organic nitrogen in fog residuals observed in the Italian Po Valley. *EGUsphere*, *2024*, 1-22. https://doi.org/10.5194/egusphere-2024-3629

**Comments from Anonymous Referee #1**

General comments:

This paper presents field data on the organic nitrogen composition of fog water and aerosol (interstitial and residual) obtained during the winter of 2022 FAIRARI campaign in the Po Valley of Italy. The measurements and analysis are novel and likely to be of great interest to the atmospheric chemistry community, and are mostly well presented in the manuscript. I have a few questions and points of confusion which I hope the authors can address.

**Reply:** We thank the reviewer for the time to assess our manuscript and the positive response. In the following we reply to each comment individually directly below the copied text from the reviewer. Only references exclusively used in this document and not part of the main manuscript are added in a separate bibliography at the end.

Specific comments:

**Comment 1:** Line 74-75: You describe the aerosol pH decrease accompanying the fog water increase, and state that the aerosol pH decrease was "mainly contributed to ammonium" – do you mean, the pH decrease was attributed to a decrease in ammonium ion concentration? Clarify, and perhaps elaborate, since this is a bit counterintuitive (that water pH would increase which aerosol pH decreases).

**Reply:** We thank the reviewer for pointing out the need for elaboration of this section. Indeed, as stated in Paglione et al. (2021) and Weber et al. (2016), total ammonium ($NH_4^+$), gas and particle phase, explains 35% of the total aerosol pH variance, with the partitioning of the semi-volatile ammonia between the gas and particle phase having a buffering effect on aerosol pH and masking the effect of decreasing concentrations of main acidic species $SO_4^{2-}$ and $NO_3^-$, which is not the case for the more dilute fog water. For detailed explanations we refer the reader to Paglione et al. (2021).The decreasing trend is a result of a decrease in precursors, including gaseous $NH_3$ during fog events. We have modified lines 71 - 76 as follows: "*Over the past three decades, numerous research studies have delved into the Po Valley fog. For example, the fog water and aerosol acidity have exhibited opposite trends. The fog water pH has increased from about 5.5 to 6.5, likely due to the decreasing trends of nitrate ($NO_3^-$) and sulfate ($SO_4^{2-}$) (Giulianelli et al., 2014). Simultaneously, the aerosol pH decreased from about 5 to 4, which was mainly contributed to total ammonium ($NH_4^+$), i.e. the buffering effect of semi-volatile ammonia, while decreasing RH and increasing temperature in the region also were major drivers (Paglione et al., 2021; Weber et al., 2016).*"

**Comment 2:** Lines 139-144: In the discussion of inlet tubing – explain why you used mixed tubing. Is there any concern about particle losses at the junctions between tubing types? And maybe list diameters also in mm.

**Reply:** The tubing going from both inlets (GCVI and WAI) were all black conductive tubing, out of convenience, i.e. simplicity in connecting lines in the tight space of the container. When possible, the tubing going to the SP-AMS was replaced by stainless steel, in order to minimize potential sampling artefacts from black conductive tubing (Timko et al., 2009). Such artefacts were observed in the mass spectra from the offline fog water analysis, during which we only used black conductive tubing (see section 2.5; lines 208 - 210). Particle losses were calculated using the Particle Loss Calculator provided by Max Planck Institute for Chemistry (von der Weiden et al., 2009). Assuming a 90°-angle for the junctions between stainless steel and black conductive tubing, within the particle diameter size range relevant for sampling with the SP-AMS (70 - 1000 nm), the calculated particle mass losses were below 5%. The inner diameters of the sampling tubes were 6.35 mm, which we added in parentheses to the manuscript.

**Comment 3:** Figure 1: I spent a long time trying to understand this diagram, so I wonder if there's a way you can help the reader get there more quickly. Maybe it would help to put red and purple lines next to each other between the 3-way switch and the SP-AMS, so it's clear that the SP-AMS switches between WAI and GCVI during fog/ non-fog? Or indicate with little arrows and labels "fog" "non-fog" which path the WAI follows in each case, at the red tee?

**Reply:** We agree with the reviewer's suggestions for the schematic figure (Fig. 1), and hope that the changes made will make it easier for the reader. We have made the following changes: We have added two lines instead of one line going from the 3-way switch to the SP-AMS, arrows on the sampling lines, and the texts "Non-fog conditions" and "Fog conditions". We also have added labels to the containers. Here is the new version:

[Figure]

**New Figure 1.** Schematic of the experimental set-up inside the two containers, including instruments relevant in this study. The two inlets in parallel, connected with a 3-way switch, allow selected instruments to change sampling inlet from the WAI to the GCVI during fog. The different colored lines represent how the instruments were connected

during ambient aerosol sampling using the WAI (red), during fog events using GCVI inlet (purple), and the interstitial aerosol sampling (green). Offline fog water analysis was performed in the laboratory (blue).

**Comment 4:** Around lines 162-164: does "dual" mode mean also BC or BC instead? The text reads like the 70 eV EI is substituted with 1064, but I think perhaps it is both simultaneous? Please clarify.

**Reply:** It is indeed both vaporizers simultaneously during the dual-vaporizer mode. Now clarified better in the text at lines 160-161 as follows: *"During dual-vaporizer mode, an intra-cavity Nd:YAG laser (1064 nm) is activated in addition to the tungsten vaporizer, which allows quantification of rBC particles."*

**Comment 5:** On line 185 you state that "During non-fog periods, both instruments measured behind the WAI." This makes me expect some general statement comparing the two BC measurements, but I don't see it.

**Reply:** We agree that a comparison between the BC measurements was missing in the manuscript. We have now added a scatterplot in the SI showing the rBC mass concentrations from the SP-AMS vs. the eBC mass concentrations from the MAAP (fog periods excluded). The BC comparison shows a strong linear correlation (Pearson correlation coefficient, r=0.91) with an intercept = 0.06. The slope = 0.70, with the SP-AMS measuring lower concentrations compared to the MAAP. Onasch et al. (2012) reported a correlation of $R^2$=0.76 between the SP-AMS and MAAP for ambient measurements, which is comparable to our correlation between the same instruments ($R^2$ = 0.83). The two instruments quantify BC by different methods, which could explain the different mass concentrations. The MAAP operates at a single wavelength of 670 nm, and one recent study by Shen et al. (2024) found that 23% of total absorption at 660 nm was due to water-soluble brown carbon (BrC) in an environment heavily influenced by biomass burning. Since the wintertime Po Valley aerosol is highly influenced by biomass burning emissions, this could potentially explain why the MAAP measured on average higher BC concentration. This should not interfere with the SP-AMS rBC, considering only $C_x$ ions are counted towards the rBC mass.
We have modified lines 186-192 as follows: *"During non-fog periods, both instruments measured behind the WAI, where they showed an overall strong linear correlation (Pearson correlation coefficient, r=0.91, see Fig. S2). The slope between the two instruments was 0.70, with the MAAP measuring on average higher mass concentrations than the SP-AMS. Since the MAAP measures at a single wavelength of 670 nm, there could be interference from e.g. brown carbon. In fact, a recent study by Shen et al. (2024) found that 23% of total absorption at 660 nm was due to water-soluble brown carbon (BrC) in an environment heavily influenced by biomass burning. Considering that the wintertime Po Valley aerosol is highly influenced by biomass burning emissions, there could be interference from BrC, to the eBC measured by MAAP."*

[Figure]

**Figure S2.** Comparison of the rBC mass concentration vs. eBC mass concentration measured with the SP-AMS and the MAAP, respectively. Data include only non-fog periods when both instruments sampled behind the WAI. The BC comparison shows a strong linear correlation (Pearson correlation coefficient, r=0.91).

**Comment 6:** Line 216: How did you quantify the concentrations of imidazole from the NMR spectrum? Here you describe, and in the SI figure, it is clear that you can identify the presence of the molecule from the existence of the peak, but how do you convert this to a mass concentration?

**Reply:** We thank the Reviewer for highlighting the lack of clarity in the presentation of the NMR methodology used. In general, the high-field NMR spectroscopy is considered accurate for molecular-level identification in complex matrices when well-resolved individual resonances match the spectra of standard compounds (Decesari et al., 2024; Tagliavini et al., 2024). Following the methods described in the cited references we always add to each sample an internal standard (Sodium 3-trimethylsilyl- (2,2,3,3-d$_4$) propionate or -TSP-d$_4$) of known concentration that allows to quantify (by proportion of their integrals with the internal standard one) every signal of the NMR spectrum. To better explain this quantification method, we have added the following to the revised text at lines 225-236: *"Briefly, the sampled aerosol was extracted with deionized ultrapure Milli-Q water in a mechanical shaker for 1 hour and aliquots of these extracts and of fog waters were dried under vacuum and re-dissolved in deuterium oxide (D$_2$O) for the analysis. The $^1$H-NMR spectra were acquired at 600 MHz in a 5 mm probe using a Varian Unity INOVA spectrometer, at the NMR facility of the Department of Industrial Chemistry (University of Bologna). $^1$H-NMR spectroscopy in protic solvents provides the speciation of hydrogen atoms bonded to carbon atoms. On the basis of their frequency shifts, the signals can be attributed to H-C containing specific functionalities (Decesari et al., 2000, 2007) and/or to specific molecules (Suzuki et al., 2001; Paglione et al., 2024). The addition of an internal standard of known concentration allows the quantification of these specific molecular tracers when well-resolved individual resonances match the spectra of standard compounds, such as 1H-imidazole, characterized by specific singlets at known chemical shifts. Sodium 3-trimethylsilyl- (2,2,3,3-d$_4$) propionate (TSP-d$_4$) was used as an internal standard by adding 50 µL of a 0.05% TSP-d$_4$ (by weight) in D$_2$O to the standard in the probe. To avoid the shifting of pH-sensitive signals, the extracts were buffered to pH 3 using a deuterated-formate/formic-acid (DCOO$^-$=HCOOH) buffer prior to the analysis."*

**Comment 7:** Around lines 241-247: You mention the moderate correlation between non-fog ambient aerosol ON and aerosol water content. Do you interpret this as indicating mainly water-soluble organic nitrogen compounds, or do you think both could be elevated during the colder episode and therefore correlated?

**Reply:** We believe the moderate correlation between non-fog ON and ALWC indicates that the ON compounds in this study mainly consist of water-soluble organic nitrogen. A considerable part of the ON is likely consisting of amino acids and amines, which generally have high hygroscopicity and have an important role in CCN activation (Chan et al., 2005; Ge et al., 2011). Furthermore, the strong correlation between the fog residual and fog water ON mass spectra indicates that they are mostly water-soluble compounds, since we can assume that the fog water samples mainly contain water-soluble compounds. We have modified the following text at lines 266-269: "*By performing a Spearman's rank correlation test, a moderate correlation was found between the non-fog ambient aerosol ON mass concentration and the ALWC ($r_s$ = 0.44, p = 0.052), consistent with previous studies (Liu et al., 2023). This moderate correlation indicates that the ions included in this ON family likely consisted of water-soluble ON compounds.*"

**Comment 8:** Figure 3: why are the data "spikier" during fog events? Maybe briefly explain in caption. Figures 3 and 4 are nice clear overview figures of the differences in composition.

**Reply:** During fog events we have less data points, so we decided to not average it over 1 hour, as we did with the non-fog data. Instead the fog residuals are averaged over 5 minutes. That is why the data looks "spikier" during fog events. Added to the caption of Figure 3: "*The ambient out-of-fog data are averaged over 1 hour, while the in-fog residual data are averaged over 5 minutes.*"

**Comment 9:** Around line 307: is the Pearson correlation coefficient you mention here just for the 1N compounds? Please clarify. Is r=0.89 the correlation coefficient for Figure S3? (all of the ON peaks) If so, mention this in the caption in the SI.

**Reply:** We agree that this needs clarification. This Pearson correlation coefficient of r=0.89 includes all ON ions ($C_xH_yN_1^+$, $C_xH_yN_2^+$ and $C_xH_yO_1N_1^+$), showing that we have a good overall agreement between the fog residuals and the fog water mass spectra. Now also included in the caption of the SI figure (Fig. S3). We have modified the following text at lines 329-331: "*Considering all ON ions identified in this study, a strong correlation was found between the fog residuals and the fog water mass spectra (Pearson correlation coefficient, r=0.89).*"

**Comment 10:** Lines 323-335: The imidazole results and comparison to NMR are very interesting indeed. On line 327 you mention they weren't observed "in PM1" samples, but later you say the NMR derived concentrations correlate well with SP-AMS. Was the SP-AMS not measuring PM1?

**Reply:** This sentence was supposed to only refer to the offline PM1 samples analyzed with the NMR spectroscopy. This has now been made more clear in the text. We changed at line 352: "*in the $PM_1$ samples*" to "*in the offline $PM_1$ samples using the $^1$H-NMR analysis.*"

**Comment 11:** Next paragraph:Do the imidazole peaks correlate at all with aerosol [NH4+]? You mention in lines 340-343 the correlation of CxHyN2+ with glyoxal being higher in the second event. Was the [NH4+] also higher?

**Reply:** We do see a similar trend when comparing $C_xH_yN_2^+$ with $NH_4^+$, as we did with glyoxal. Over the full measurement period, no significant linear correlation was found between $NH_4^+$ and $C_xH_yN_2^+$ in the ambient (r=0.26) or in the fog residuals (r=0.29), similarly to between the glyoxal fragments and $C_xH_yN_2^+$. For the second intensive fog period (27/03 - 30/03), the correlation increased both in the ambient aerosol (r=0.64) and in the fog residuals (r=0.67). The $NH_4^+$ mass concentration was slightly lower during the second period, however, $NH_4^+$ contributed a similar mass fraction to the aerosol composition in the two periods, both in fog residuals and ambient $PM_1$. $[NH_3](g)$ was similar during both periods, while the $NO_3^-$ concentration was considerably lower, which should mean that there was more $NH_3$ / $NH_4^+$ available for the formation of imidazoles during the second intensive fog period.

We made changes in this paragraph accordingly, starting at line 363: "*Previous laboratory studies have found evidence of aqueous-phase formation of ON compounds (such as imidazoles) from uptake of glyoxal with amino acids (De Haan et al., 2009) or with ammonium (Galloway et al., 2009). Glyoxal, an important precursor for aqSOA, typically fragments into $C_2H_2O_2^+$, $C_2H_2^+$, $CH_2O_2^+$ from electron impact ionization (Paglione et al., 2020). Over the full measurement period, no significant linear correlation (Pearson's correlation coefficient) was found between the $C_xH_yN_2^+$ ions and the glyoxal fragments in the ambient aerosol (r=0.38) and in the fog residuals (r=0.37). Similarly, the $C_xH_yN_2^+$ ions did not show a correlation with $NH_4^+$ in the ambient aerosol (r=0.26) or in the fog residuals (r=0.29). However, considering only the second intensive fog period (March 27-30), the $C_xH_yN_2^+$ ions were much more abundant in the ambient aerosol and slightly more abundant in the fog residuals. During this period, stronger correlations were found in both the ambient aerosol (r=0.65 and r=0.64 with glyoxal and $NH_4^+$, respectively) and in the fog residuals (r=0.70 and r=0.67 with glyoxal and $NH_4^+$, respectively). Furthermore, the high concentration of $NO_3^-$ during the first intensive fog period suggests that $HNO_3$ was a large sink for $NH_3$. During the second intensive fog period, however, the $NO_3^-$ concentration was considerably lower, which could mean more $NH_3$ / $NH_4^+$ was available for imidazole formation. Throughout the campaign, the concentration of gas-phase $NH_3$ was relatively similar, fluctuating between 10 and 30 ppb (see Fig. S6). No considerable difference was found between the first (17.3 ppb) and second (15.9 ppb) intensive fog period for $NH_3$. While data were missing for more than half of the first period, it was found reasonable to assume that the $NH_3$ should follow a similar trend over those days. Together with the more OA-dominated aerosol, this could have enhanced the formation of imidazoles through glyoxal uptake with ammonium, suggested both by the increased abundance of imidazole fragments and the stronger correlations with glyoxal and $NH_4^+$ during the second intensive fog period. Meanwhile, good linear correlations were found between the $C_xH_yN_1^+$ ions and glyoxal fragments throughout the campaign (ambient r=0.80 and fog residuals r=0.84). Since glyoxal is a precursor of aqSOA, this suggests that a considerable part of the ON compounds either are formed, or partition into the aerosol, during similar conditions as for aqSOA formation.*" Furthermore, we have included a figure in the SI showing the concentration of $NH_3$. New Fig S6:

[Figure]

**Figure S6.** 5-minute averaged concentration of gas-phase ammonia ($NH_3$). Fog events are indicated by the gray areas.

**Comment 12:** At the end of the paragraph (lines 344-345) you mention a stronger linear correlation of CxHyN1+ with glyoxal. How do you interpret this?

**Reply:** The stronger correlation between $C_xH_yN_1^+$ ions and glyoxal fragments indicate similar temporal behavior of formation and removal processes. Since it has previously been established that glyoxal is a good tracer for aqSOA formation, we can say that the strong correlation indicates that the $C_xH_yN_1^+$ ions increase in the aerosol during similar conditions as for aqSOA formation.

We changed the sentence at lines 380-383 as follows: *"Meanwhile, good linear correlations were found between the $C_xH_yN_1^+$ ions and glyoxal fragments throughout the campaign (ambient r=0.80 and fog residuals r=0.84). Since glyoxal is a precursor of aqSOA, this suggests that a considerable part of the ON compounds either are formed, or partition into the aerosol, during similar conditions as for aqSOA formation."*

**Comment 13:** Figure 6: I find the symbology on this figure confusing. Why are only the fog residuals shown as box & whisker plots and the others as bar charts behind? I think this would be much easier to read with all data as box & whiskers. The overlaying is not helpful, and you could colors to indicate which axis each corresponds to. Do you have ambient PM1 data also from the SP-AMS? This would provide another point of comparison.

**Reply:** We thank the reviewer for this comment. We agree that Figure 6 needs to be improved. It would have been possible to present all data as box & whiskers, however, the NMR results would give an indication of the repeatability, while the SP-AMS measurements indicate the variability during the fog event. For this reason, we instead decided to display all data as markers with error bars representing one standard deviation. Furthermore, we followed your advice to include the daily average $C_xH_yN_2^+$ mass concentrations from the ambient aerosol, and to color the axes according to the color of the data it represents. All y-axes were also changed to display the mass concentration in ng m$^{-3}$ instead of μg m$^{-3}$.

[Figure]

**Figure 6.** Average mass concentration of $C_xH_yN_2^+$ measured by SP-AMS, and 1H-imidazole by $^1$H-NMR spectroscopy for each fog event. Panel (a) shows the daily average sub-micron ambient $C_xH_yN_2^+$ (red, left y-axis) and the 1H-imidazole from offline filter samples (light blue, right y-axis). Panel (b) includes the fog residual $C_xH_yN_2^+$ (light green, left y-axis), fog water $C_xH_yN_2^+$ (dark green, right y-axis), and fog water 1H-imidazole (purple, right y-axis). The fog water 1H-imidazole concentration in mass per m$^3$ of air was obtained by multiplying its concentration in fog water

with the liquid water content of the air, measured by a Particulate Volume Monitor. The fog water $C_xH_yN_2^+$ concentration is displayed in mass per m$^3$ of nebulized air, and is thus only used to look at the trend. Each symbol represents the mean, while the error bars represent one standard deviation.

**Comment 14:** Line 364: clarify that the lower temp and higher RH occurred during the first period. Do I correctly infer that this means you expect the nitrate to be more semivolatile than the OA mix?

**Reply:** We indeed intended to say that the lower temperature and higher RH occurred during the first period. On average throughout the campaign, the $NO_3^-$ aerosol seems to show slightly larger variations due to the temperature, indicating that it is more semi-volatile than the OA mix. Now clarified better in the text at lines 402-405 as follows: "*This was likely due to the on average lower temperature and simultaneously higher RH during the first period, which allowed more gas-particle partitioning of especially inorganic compounds. On average over the measurement campaign, $NO_3^-$ showed slightly larger variation due to temperature, indicating that it is more semi-volatile than the OA.*"

**Comment 15:** Around line 375-377: Why would preferred partitioning of amines into the aqueous phase result in a strong correlation with glyoxal? If they have different sources, the gas-phase concentrations are not necessarily tracking.

**Reply:** We thank the reviewer for this comment. We agree that this sentence was not clearly phrased. Paragraph at line 414 has now been changed to: "*A substantial enhancement of ON was found in both fog residuals and fog water samples, which also proved to be highly correlated. The fog-enhanced ON consisted of several $C_xH_yN_1^+$ ions, likely originating from various amino compounds, while a few ions could be identified as primary and secondary amines. The increased abundance of these compounds in the fog, while also showing good correlation with glyoxal fragments and moderate correlation with ALWC, suggests that a considerable amount of the ON observed in the particle phase are either produced by aqueous-phase chemical reactions, or partitioning between the gas and aqueous phase.*"

**Comment 16:** Around line 381-382: Could the elevated CxHyN2+ during period 2 also be related to that period having dominant OA in the aerosol rather than NO3-? Does elevated NO3- provide another "sink" for NH4+ other than formation of imidazole? Again, I'd be interested to hear more about correlations with NH4+ as well.

**Reply:** We thank the Reviewer for the comment, and agree that the discussion on why we see elevated $C_xH_yN_2^+$ concentrations in the 2nd period was also missing in the conclusions. Again, we think it is a good hypothesis that $NO_3^-$ is one of the main "sinks" for $NH_4^+$ in the 1st period, and is substantially reduced during the 2nd period. See also our reply to comment 11, which is on the same topic. Consequently, the excess $NH_3$ / $NH_4^+$ could be available for reactions with glyoxal, to form imidazoles. The paragraph at line 421 was rewritten to: "*The $C_xH_yN_2^+$ ions, mainly present in the fog water, suggest formation of imidazoles in the fog droplets. The related fragments linked to imidazoles have been confirmed by [1]H-NMR spectroscopy, and thus can now be used as tracer fragments. We hypothesize the reason they were less present in the fog residuals was that they were more concentrated in the large fog droplets, not sampled by the GCVI inlet. In the second intensive fog period, these $C_xH_yN_2^+$ ions were substantially elevated in the fog residuals, and even persisted in the aerosol phase after fog dissipation, which was not observed during the first intensive fog period. During this period, the $C_xH_yN_2^+$ ions showed strong positive correlation with both glyoxal fragments and $NH_4^+$, which was not found during other periods. Compared to the first intensive fog period, the aerosol mass consisted of a lower $NO_3^-$ concentration, while the fraction of OA was considerably higher, and $NH_4^+$ stayed relatively constant. This could explain the elevated mass concentration of $C_xH_yN_2^+$ ions, since more $NH_3$ / $NH_4^+$ was available for aqueous reactions with glyoxal, or*

*other α-dicarbonyls, to form imidazoles. Since the formation of 1H-imidazole is irreversible, even small amounts could have an effect on the aerosol optical properties. This shows the importance of fog processing from both an air quality and climate perspective. The results from this study improve our understanding on fog processing, and demonstrate that fogs and clouds are an important medium for aqueous production of certain nitrogen-containing organic aerosol in the atmosphere."*

**Comment 17:** SI: In the caption to figure S1, I suggest to remind the reader briefly the nature of the correction applied.

**Reply:** We added the following text to the caption of Fig. S1: "*The discrepancy between the two instruments was due to an underestimated ionization efficiency for the SP-AMS, resulting in an overestimated mass concentration, likely due to a bad calibration setup. Therefore, a correction factor of 0.51 was derived by comparison with the interstitial AMS, which was validated by ion-chromatography on parallel PM$_1$ filter samples.*" We also updated the figure to give some additional information, and have a matching style to Fig. S2, included here:

[Figure]

**Figure S1.** Comparison of the NR-PM$_1$ mass concentration between the SP-AMS and the interstitial AMS, before correction (red) was applied and after applying the correction (blue). The discrepancy between the two instruments was due to an underestimated ionization efficiency for the SP-AMS, resulting in an overestimated mass concentration, likely due to a bad calibration setup. Therefore, a correction factor of 0.51 was derived by comparison with the interstitial AMS, which was validated by ion-chromatography on parallel PM$_1$ filter samples.

**Technical corrections:**

**Comment 18:** Line 13 and elsewhere: when you use "1H-NMR" to refer to proton NMR, the 1 should be superscripted

**Reply:** Corrected.

**Comment 19:** Line 14: "medium" -> "media"

**Reply:** Corrected.

**Comment 20:** Line 27: suggest "understood" rather than "quantified"

**Reply:** Corrected at line 38.

**Comment 21:** Line 37: suggest "aerosol precursors" rather than "air pollutants"

**Reply:** This sentence was removed. See comment 36 from Reviewer 2.

**Comment 22:** Line 51: "atmospheric reactive nitrogen"

**Reply:** Corrected.

**Comment 23:** Line 88-89: suggest "The analysis of aerosol-fog interactions .. been restricted to offline chemical …" (make more general)

**Reply:** Corrected. The sentence at lines 88-89 now reads: *"The analysis of aerosol-fog interactions has so far been restricted to offline chemical analysis of fog water or aerosol chemical composition, or to in-situ analysis of interstitial aerosol particles."*

**Comment 24:** Line 98: "The findings from Kim … in the ambient aerosol, and this work extends this quantification to fog water and fog residuals." (however doesn't seem to fit here)

**Reply:** We agree that this sentence did not fit as it was. Now rewritten at lines 96-98 to: "*Previous studies performing online quantification of ON exist (e.g. Kim et al., 2019; Ge et al., 2024; Graeffe et al., 2022) but are relatively few compared to studies on other OA species. This work presents an effort to highlight particulate ON existing in fog water and fog residuals.*"

**Comment 25:** Line 124: remove "around"?

**Reply:** Corrected as suggested.

**Comment 26:** Line 134-135: suggest "Karlsson et al., 2021), therefore the sampled fog residual masses should be considered a lower limit of the real ambient values."

**Reply:** Sentences at lines 131-133 were corrected to: "It should be noted that we did not account for the droplet sampling efficiency of the GCVI, as suggested by Shingler et al. (2012) and Karlsson et al. (2021). Therefore, the sampled fog residual masses should be considered a lower limit of the real ambient values."

**Comment 27:** Line 135: "Visibility was measurement by a Belfort Instrument sensor (Model 6400)."

**Reply:** Sentence at line 134 was corrected to: "*Visibility was measured with a Visibility Sensor Model 6400 (Belfort Instrument, USA).*"

**Comment 28:** Line 176: "calibration"

**Reply:** Corrected.

**Comment 29:** Line 181: does "within this range" mean "below m/z 106"? Or do you want to cite a m/z range here?

**Reply:** Sentence at line 181 rewritten to: *"The W-mode was set to scan up to m/z 106, which means that only ON ions up to this m/z were included during the peak fitting."*

**Comment 30:** Line 190: on Ca and Mg the 2 should be part of the cation superscript

**Reply:** Corrected.

**Comment 31:** Line 340-341: suggest "(March 27-30), when $C_xH_yN_2^+$ ions were much more abundant in the ambient aerosol and fog residuals, a stronger linear correlation …"

**Reply:** Corrected.

**Comment 32:** Line 347: suggest "in the online fog residual measurements"

**Reply:** Corrected.

**Comment 33:** Line 366: suggest "comparison" > "contrast"

**Reply:** Corrected.

**Comment 34:** Line 379: suggest reordering: "The related fragments linked to imidazole have been confirmed by [1]H-NMR spectroscopy, and thus can now be used as tracer fragments."

**Reply:** Corrected

**Comments from Anonymous Referee #2**

General comments:

The researchers performed online measurements of fog residuals and ambient aerosols, as well as offline analysis of fog water in the Po Valley, with a focus on the organic nitrogen components. It was found that both the fog droplet residuals and fog water had enhancements of amino compounds and imidazoles. The manuscript is well written and informs an interesting topic area. I only have a few comments:

**Reply:** We thank the reviewer for the positive assessment of our manuscript. In the following we reply to each comment individually directly below the copied text from the reviewer. Only references exclusively used in this document and not part of the main manuscript are added in a separate bibliography at the end.

Specific comments:

**Comment 35:** Do the authors have any comments on the primary sources of ON and relevant formation pathways of imidizoles in the Po Valley? Line 52 mentions vehicle exhaust, biomass burning and agriculture as possible sources, and at line 312 the authors seem to dismiss local traffic sources as a

source due to differences from Saarikoski et al (2012). A sentence or two in the conclusions will help showcase the relevance to other geographic regions.

**Reply:** We thank the reviewer for the question about sources and formation of ON and imidazoles in the Po Valley. The $C_xH_yN_1^+$ ions comprise the largest part of the ON we observe during this study, and from those ions we were able to identify a few specific aliphatic amines. These amines have a variety of potential sources, such as animal husbandry, biomass burning, industry, traffic, and even pesticides. Since we did not observe a typical daily pattern of traffic, we exclude local vehicle exhaust as a major source of the ON. We know from previous studies from the Po Valley that biomass burning is a major source of both primary and secondary OA in the winter, while agriculture and animal husbandry are major sources of VOCs and ammonia (Gilardoni et al., 2016, Saarikoski et al., 2012, Paglione et al., 2020). ON species, such as amines, have been found to contribute to aerosol mass both through gas-phase reactions (oxidation reactions with OH, $NO_x$, or $O_3$), as well as acid-base reactions (Ge et al., 2011). We believe that the ON observed in this study likely originates from various sources, where biomass burning, agriculture, and animal husbandry are the main sources. For imidazoles, it is possible that a major formation pathway is the uptake of either amines or glyoxal by ammonium in the fog droplets / deliquesced particles (see also our replies to comment 11 and comment 13 from reviewer 1). At the same time, we cannot exclude the possibility that the enhanced imidazole mass in the fog droplets could arise from dissolution of the gaseous fraction into the fog droplets / fog water. However, the strong correlations with glyoxal and $NH_4^+$ during the second period, where the mass concentration is much higher in the ambient aerosol and the fog residuals, indicate that it forms from this reaction in the aqueous phase. For future work, performing a positive matrix factorization analysis to investigate the potential sources, would certainly be useful in order to investigate the potential sources of ON compounds. We have added the following to the revised manuscript at line. 416 : *"These ON compounds are likely products from biomass burning, agriculture, and animal husbandry emissions."*

**Comment 36:** Line 46: "Inorganic aerosol species also contribute to aqSOA. Nitrate (NO−3) is becoming more important as an aerosol component in many areas, due the reduction of other air pollutants (e.g. SO2), or where high levels of excess NH3 exist in the atmosphere (Bauer et al., 2007)." Is this sentence stating that high levels of nitrate can enhance aqSOA formation, or that inorganic nitrate is an important PM component formed in the aqueous phase. If the latter, the first sentence should be revised as nitrate is not aqSOA.

**Reply:** We agree with the reviewer that this sentence was incorrect. We thus decided to remove that sentence, and the sentence after, at line 46 because they did not fit with the rest of the paragraph: *"Inorganic aerosol species also contribute to aqSOA. Nitrate ($NO_3^-$) is becoming more important as an aerosol component in many areas, due to the reduction of other air pollutants (e.g. $SO_2$), or where high levels of excess $NH_3$ exist in the atmosphere (Bauer et al., 2007)."*

**Comment 37:** Line 191: Were these ions measured using the AMS or the offline IC that was briefly mentioned previously. If it was the AMS: how were K+, Ca+, Mg2+ and Na+ treated

**Reply:** We understand this was not stated in the manuscript and thank the reviewer for the comment. The ions $K^+$, $Ca^{2+}$, $Mg^{2+}$, and $Na^+$ were measured using offline ion-chromatography on PM1 filter samples. Sentence at lines 198-201 now reads: *"In addition, the aerosol liquid water content (ALWC) was predicted using the aerosol inorganic species; $NO_3^-$ , $SO_4^{2-}$ , $NH_4^+$, and $Cl^-$ measured by HR-ToF-AMS, as well as $K^+$, $Ca^{2+}$, $Mg^{2+}$, and $Na^+$ from offline ion-chromatography on $PM_1$ filter samples, as inputs in ISORROPIA-II model (Fountoukis and Nenes, 2007)."*

**Comment 38:** Line 215: Does the NMR method differentiate between different substituted imidazoles?

**Reply:** With the NMR method we only analyzed 1H-imidazole. Other substituted imidazoles, such as 2-Methylimidazole, could have singlets in the spectral region between 7 and 7.5ppm (actually visible in most of the fog spectra in Fig. S4) but for the moment they are not unambiguously identifiable. The inclusion of more N-heterocyclic compounds would be very interesting for future work.

**Comment 39:** Figure 6: I find this figure hard to interpret. Is there another way to present these results such that it is clearer which bars correspond to which axis? I also find the simultaneous use of the box plot and bars distracting.

**Reply:** We thank the reviewer for this comment. We refer to our reply to comment 13 from reviewer 1 for the exact changes made to Fig. 6.

**Editor comments:**

**Comment 1:** Line 192: correct the typos for Ca2+ and Mg2+;

**Reply:** Corrected.

**Comment 2:** Line 426: add the last access date

**Reply:** Corrected.

**Further changes made to manuscript:**

1. The collection efficiency of the fog water collector (FWC) was incorrect, due to a typo in a previous paper. It has now been fixed, along with related texts.
At line 206 (preprint line: 198) text was included as follows: *"and had a 50% collection efficiency for each individual string at approximately 3 $\mu$m droplet radius (Fuzzi et al., 1997)."*
At lines 318-320 (preprint lines: 296-298) text was modified as follows: *"Another hypothesis is that the chemical composition was not consistent over all droplet sizes, which could explain the disparity between the fog water and residuals, due to the different droplet size range of the two sampling methods, (i.e. FWC and GCVI)."*
At lines 384-395 (preprint lines: 346-351) the whole paragraph was modified as follows: *"For imidazoles, little is known about their physicochemical properties, such as volatility. According to UK REACH (2018), 1H-imidazole has a vapor pressure of 3.27 x 10$^{-10}$ Pa, which would mean that it has a low volatility. A more recent study estimated imidazoles in general to have a vapor pressure of 0.2 Pa or more, which instead would mean that they are highly volatile (Amugoda and Davies, 2025). However, they show that imidazoles stay in the particle phase, despite the high volatility, when they are internally mixed with inorganic salts. Therefore, imidazoles are expected to remain in the particle phase after droplet evaporation in the different inlets and drying of the aerosol. De Haan et al. (2011) found formation of imidazoles, from nitrogen-containing species (i.e. ammonium and amines) and glyoxal, during simulated evaporation of droplets. Ervens et al. (2011) summarized that these reactions likely occur in the evaporating droplets, when glyoxal has dissolved in the water and, simultaneously, the evaporation of water results in increasing concentrations of the nitrogen-containing compounds. A possible explanation why the $C_xH_yN_2^+$ ions were reduced in the online fog residual measurements, compared to fog water*

*samples, could be that they were more concentrated in the smaller evaporating fog droplets, which were sampled more efficiently by the FWC compared to the GCVI.*"

At lines 412-413 (preprint lines: ) text was modified as follows: *"Thus, we hypothesize this discrepancy could be due to different sampling methods and that the chemical composition could be dissimilar in larger versus smaller fog droplets."*

2. Reference "Neuberger et al., 2024, in press" now published. Fixed in text and reference list of revised manuscript.

3. Included Table S1. in the supplementary showing the exact start and end times included for each fog event included in this study. Referenced in the main text at line 240.

**References**

Chan, M. N., Choi, M. Y., Ng, N. L., & Chan, C. K. (2005). Hygroscopicity of water-soluble organic compounds in atmospheric aerosols: Amino acids and biomass burning derived organic species. *Environmental science & technology*, *39*(6), 1555-1562.

Decesari, S., Paglione, M., Mazzanti, A., & Tagliavini, E. (2024). NMR spectroscopic applications to atmospheric organic aerosol analysis–Part 1: A critical review of data source and analysis, potentialities and limitations. *TrAC Trends in Analytical Chemistry*, *171*, 117516.

Ge, X., Wexler, A. S., & Clegg, S. L. (2011). Atmospheric amines–Part I. A review. *Atmospheric Environment*, *45*(3), 524-546.

Gilardoni, S., Massoli, P., Paglione, M., Giulianelli, L., Carbone, C., Rinaldi, M., Decesari, S., Sandrini, S., Costabile, F., Gobbi, G. P., Pietrogrande, M. C., Visentin, M., Scotto, F., Fuzzi, S., & Facchini, M. C. (2016). Direct observation of aqueous secondary organic aerosol from biomass-burning emissions. *Proceedings of the National Academy of Sciences*, *113*(36), 10013-10018.

Giulianelli, L., Gilardoni, S., Tarozzi, L., Rinaldi, M., Decesari, S., Carbone, C., Facchini, M. C., & Fuzzi, S. (2014). Fog occurrence and chemical composition in the Po valley over the last twenty years. *Atmospheric Environment*, *98*, 394-401.

Onasch, T. B., Trimborn, A., Fortner, E. C., Jayne, J. T., Kok, G. L., Williams, L. R., Davidovits, P., & Worsnop, D. R. (2012). Soot particle aerosol mass spectrometer: development, validation, and initial application. *Aerosol Science and Technology*, *46*(7), 804-817.

Paglione, M., Gilardoni, S., Rinaldi, M., Decesari, S., Zanca, N., Sandrini, S., Giulianelli, L., Bacco, D., Ferrari, S., Poluzzi, V., Scotto, F., Trentini, A., Poulain, L., Herrmann, H., Wiedensohler, A., Canonaco, F., Prévôt, A. S. H., Massoli, P., Carbone, C., Facchini, M. C., & Fuzzi, S. (2020). The impact of biomass burning and aqueous-phase processing on air quality: a multi-year source apportionment study in the Po Valley, Italy. *Atmospheric Chemistry and Physics*, *20*(3), 1233-1254.

Paglione, M., Decesari, S., Rinaldi, M., Tarozzi, L., Manarini, F., Gilardoni, S., Facchini, M. C., Fuzzi, S., Bacco, D., Trentini, A., Pandis, S. N., & Nenes, A. (2021). Historical changes in seasonal aerosol acidity in the Po Valley (Italy) as inferred from fog water and aerosol measurements. *Environmental science & technology*, *55*(11), 7307-7315.

Saarikoski, S., Carbone, S., Decesari, S., Giulianelli, L., Angelini, F., Canagaratna, M., Ng, N. L., Trimborn, A., Facchini, M. C., Fuzzi, S., Hillamo, R., & Worsnop, D. (2012). Chemical characterization of springtime submicrometer aerosol in Po Valley, Italy. *Atmospheric Chemistry and Physics*, *12*(18), 8401-8421.

Shen, Y., Pokhrel, R. P., Sullivan, A. P., Levin, E. J., Garofalo, L. A., Farmer, D. K., Permar, W., Hu, L., Toohey, D. W., Campos, T., Fischer, E. V., & Murphy, S. M. (2024). Understanding the mechanism and importance of brown carbon bleaching across the visible spectrum in biomass burning plumes from the WE-CAN campaign. *Atmospheric Chemistry and Physics*, *24*(22), 12881-12901.

Tagliavini, E., Decesari, S., Paglione, M., & Mazzanti, A. (2024). NMR spectroscopic applications to atmospheric organic aerosol analysis–Part 2: A review of existing methodologies and perspectives. *TrAC Trends in Analytical Chemistry*, 117595.

Timko, M. T., Yu, Z., Kroll, J., Jayne, J. T., Worsnop, D. R., Miake-Lye, R. C., Onasch, T. B., Liscinsky, D., Kirchstetter, T. W., Destaillats, H., Holder, A. L., Smith, J. D., & Wilson, K. R. (2009). Sampling artifacts from conductive silicone tubing. *Aerosol Science and Technology*, *43*(9), 855-865.

von der Weiden, S.-L., Drewnick, F., & Borrmann, S. (2021). Particle Loss Calculator – a new software tool for the assessment of the performance of aerosol inlet systems. *Atmos. Meas. Tech., 2,* 479–494.

Weber, R. J., Guo, H., Russell, A. G., & Nenes, A. (2016). High aerosol acidity despite declining atmospheric sulfate concentrations over the past 15 years. *Nature Geoscience*, *9*(4), 282-285.